# Spatial validation reveals poor predictive performance of large-scale ecological mapping models

Pierre Ploton [1✉], Frédéric Mortier [2,3], Maxime Réjou-Méchain[1], Nicolas Barbier [1], Nicolas Picard[4], Vivien Rossi [5], Carsten Dormann [6], Guillaume Cornu [2,3], Gaëlle Viennois[1], Nicolas Bayol[7], Alexei Lyapustin[8], Sylvie Gourlet-Fleury [2,3] & Raphaël Pélissier [1]

Mapping aboveground forest biomass is central for assessing the global carbon balance. However, current large-scale maps show strong disparities, despite good validation statistics of their underlying models. Here, we attribute this contradiction to a flaw in the validation methods, which ignore spatial autocorrelation (SAC) in data, leading to overoptimistic assessment of model predictive power. To illustrate this issue, we reproduce the approach of large-scale mapping studies using a massive forest inventory dataset of 11.8 million trees in central Africa to train and validate a random forest model based on multispectral and environmental variables. A standard nonspatial validation method suggests that the model predicts more than half of the forest biomass variation, while spatial validation methods accounting for SAC reveal quasi-null predictive power. This study underscores how a common practice in big data mapping studies shows an apparent high predictive power, even when predictors have poor relationships with the ecological variable of interest, thus possibly leading to erroneous maps and interpretations.

[1] AMAP, Univ Montpellier, IRD, CNRS, INRAE, CIRAD, Montpellier, France. [2] CIRAD, UPR Forêts et Sociétés, F-34398 Montpellier, France. [3] Université de Montpellier, F-34000 Montpellier, France. [4] Via della Sforzesca 1, 00185 Rome, Italy. [5] CIRAD, UPR Forêts et Sociétés, Yaoundé, Cameroon. [6] Biometry and Environmental System Analysis, University of Freiburg, Freiburg im Breisgau, Germany. [7] Forêt Ressources Management Ingénierie, 34130 Mauguio, Grand Montpellier, France. [8] NASA Goddard Space Flight Center, Greenbelt, Maryland 20771, USA. ✉email: p.ploton@gmail.com

Tropical forests have a key role in Earth's carbon cycle, but estimations of stocks and fluxes in these ecosystems remain limited by large uncertainties[1]. In the last decade, space-borne remote sensing (RS) has emerged as a promising way to generate transparent and globally consistent spatiotemporal syntheses of aboveground forest carbon stocks at pantropical[2,3] or global scales[4]. In particular, two reference pantropical carbon-density maps have been produced using a combination of environmental and RS predictors[2,3]. These maps have been used in high-ranking studies to estimate greenhouse gas emissions[5], to assess the relationships between forest carbon and biodiversity[6], climate[7], and land management[8–10], or even to evaluate the sensitivity of new space-borne sensors to aboveground biomass (AGB)[4,11,12]. It is therefore very concerning that despite report-edly high predictive power of their underlying prediction models, these reference maps show major disagreements[13–15] and poorly correlate with higher-quality AGB maps produced at smaller scales[16]. For instance, in the world's second-largest tropical forest block in central Africa, the two maps show strikingly opposite regional patterns of AGB variation[14]. A number of reasons have been evoked to explain these discrepancies, including the con-tamination of RS optical imagery by clouds or the use of different calibration data sets[17]. We, however, argue that a first-order methodological drawback in the validation scheme of mapping models, which ignores spatial dependence in the raw data, masks overfitting and leads to highly optimistic evaluations of predictive power. Here, our objective is to draw attention to this overlooked yet critical methodological issue in the large-scale mapping of any ecological variable, reproducing recent efforts in forest biomass mapping as a study case.

To provide evidence for this issue, we replicated the general approach for such mapping exercises, which consists of training a prediction model on a discrete sample of reference AGB data to project the estimations outside sampling areas, based on a set of external, supposedly predictive variables. For that purpose, we used a massive set of reference AGB data derived from more than 190,000 management forest inventory plots spread over five countries in central Africa[18]. This unique data set, acquired between the early 2000s and 2010s, represents a cumulative sampling area of nearly 100,000 ha and c. 11.8 million trees that were measured and identified. Inventory data were analyzed through a standardized computation scheme to provide AGB estimates at the plot level, which were then aggregated into c. 60,000 1-km pixels[18] (Fig. 1). The final set of reference AGB pixels spans the area from the Atlantic coasts of Gabon and Cameroon to the Democratic Republic of Congo inlands and thus covers wide regional gradients in terms of forest composition and climate[19]. These AGB pixels constitute our model training data set, having the same role here as AGB estimations derived from Geoscience Laser Altimeter System (GLAS) data in other AGB mapping studies[2,3].

The wall-to-wall mapping step from these discrete sampling points, however, poses the greatest challenge, as no single external variable provides both exhaustive coverage of the Earth's surface and strong sensitivity to dense forest AGB variation. For instance, a plethora of environmental (e.g., climate, topography, soil types) variables is available at a global scale, but environmental control over forest AGB is either weak[20] or highly context-dependent[16]. Similarly, current space-borne sensors that provide wall-to-wall measurements, such as radar scatterometers or multispectral imagers, usually show nonlinear sensitivity to biophysical forest properties, becoming uninformative beyond a saturation threshold of ~100 Mg ha$^{-1}$ of AGB for MODIS and ALOS[21], for instance, which is far below the average AGB of carbon-rich tropical for-ests[22]. However, the advent of machine-learning (ML) techniques has apparently allowed a quantitative leap in the predictive power

of AGB models based on these sensors. Several studies indeed combined environmental layers with RS data (such as MODIS[2,3] or QuikSCAT[2]) to produce supposedly highly predictive mapping models immune to saturation and inflation of prediction errors up to AGB levels as high as 500 Mg ha$^{-1}$. Despite ongoing debate on the relationship (or lack thereof) relating dense forest reflectance to AGB[23], it has even been claimed that subtle annual changes in AGB associated with forest degradation and growth could be reliably monitored by MODIS-based ML models[24].

Following the same approach, we thus used the random forest (RF) algorithm[25] to model dense African forest AGB derived from the abovementioned inventory data[18] using environmental and MODIS variables as predictors. MODIS data corresponded to a 10-year composite (2000–2010) of the latest collection (MAIAC product[26]), which notably includes improved atmospheric cor-rections. Environmental conditions were characterized using cli-mate variables from the WorldClim-2 database[27] and topographic variables derived from SRTM data.

In mapping studies, a classical procedure is to evaluate model performance and associated errors by randomly selecting a number (e.g., 10%) of test observations (here, 1-km pixels of "observed" AGB) that are set aside at the model calibration stage and only used to quantify model prediction error (validation step). This procedure, used in pantropical carbon mapping stu-dies[2,3], can be iterated $K$ times with different test and training sets for model cross-validation (hereafter random $K$-fold CV). Instinctively, assessing model prediction error associated with "new" pixels makes perfect sense since they mimic unsampled pixels that represent the vast majority of most maps. However, the random selection of test observations does not warrant independence from training observations when dependence structures exist in the data, i.e., when observations close to each other tend to have similar characteristics. This phenomenon, known as spatial autocorrelation, indeed has two major unde-sirable consequences for statistical modeling. A first well-known issue is that it may lead to autocorrelated model residuals, for instance, when the model lacks an important explanatory variable that spatially structures the response. Spatial autocorrelation in the residuals is problematic because it violates the assumption of error independence shared by most standard statistical proce-dures, which often leads to biases in model parameter estimates and optimistic parameter standard errors[28,29]. A second and separate issue, which has received much less attention by ecolo-gists, is that spatial autocorrelation in the raw data can invalidate model validation approaches because an observation point cannot serve as spatially independent validation of nearby training data points[30].

For instance, in our case, an empirical variogram shows that at a 1-km spatial resolution, forest AGB presents a significant spatial correlation up to c. 120 km (Fig. 2a). This spatial autocorrelation can notably be observed in Fig. 1, where patches of relatively homogeneous AGB values are visible. Climate, topographic and optical variables considered to model AGB variations are also strongly structured in the geographical space with autocorrelation ranges on the order of c. 250–500 km (Fig. 2b). Given the rela-tively high sampling intensity (and resulting proximity) of forest inventory data (Fig. 3) and the long range of spatial auto-correlation in AGB data (>100 km), it is obvious that any given randomly selected test pixel will not be independent from a large number of neighboring pixels (Fig. 3c), thereby violating the core hypothesis of model validation (i.e., the independence between training and test sets). Again, this second issue is separate from the first one, in that, as shown in the following, even a model with no apparent spatial autocorrelation in the residuals cannot be validated correctly for its predictive ability when the training and test data sets are spatially dependent.

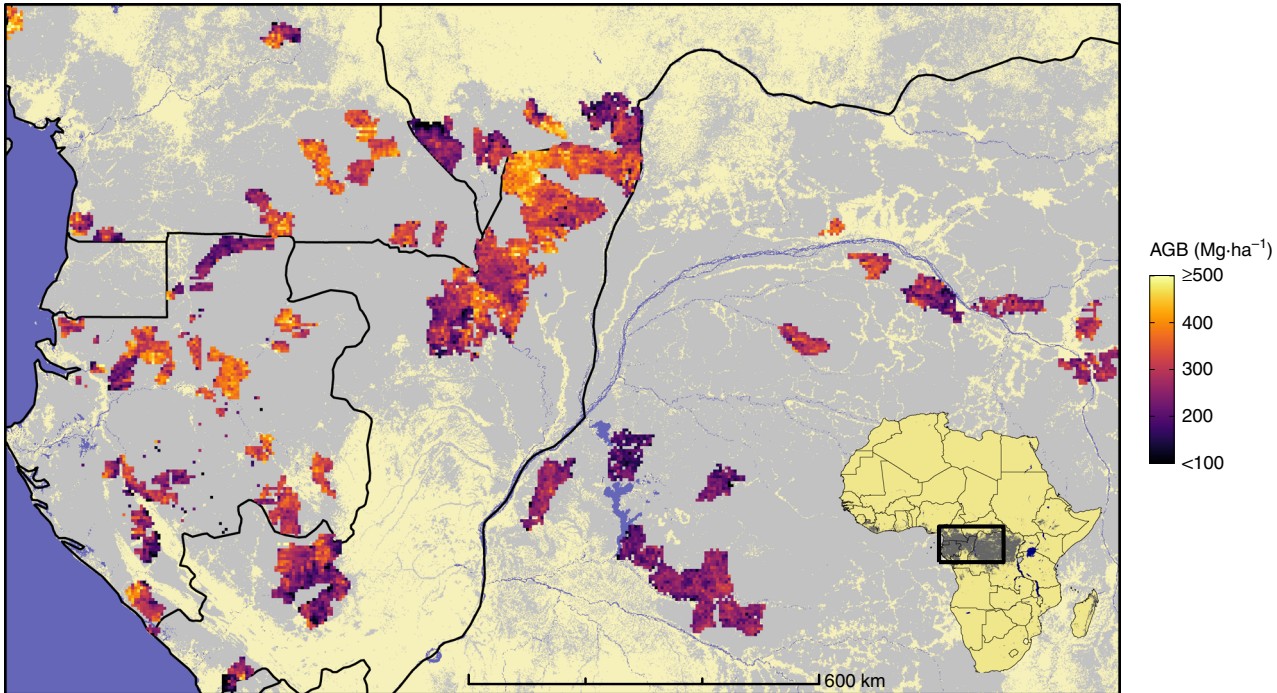

**Fig. 1 Overview of the study area and field data distribution.** Reference forest AGB estimations derived from field inventories, aggregated into 5-km pixels (for visual purposes), are depicted in a magenta-to-yellow color gradient and are superimposed over the spatial distribution of moist forests[61] (gray) and country borders (black).

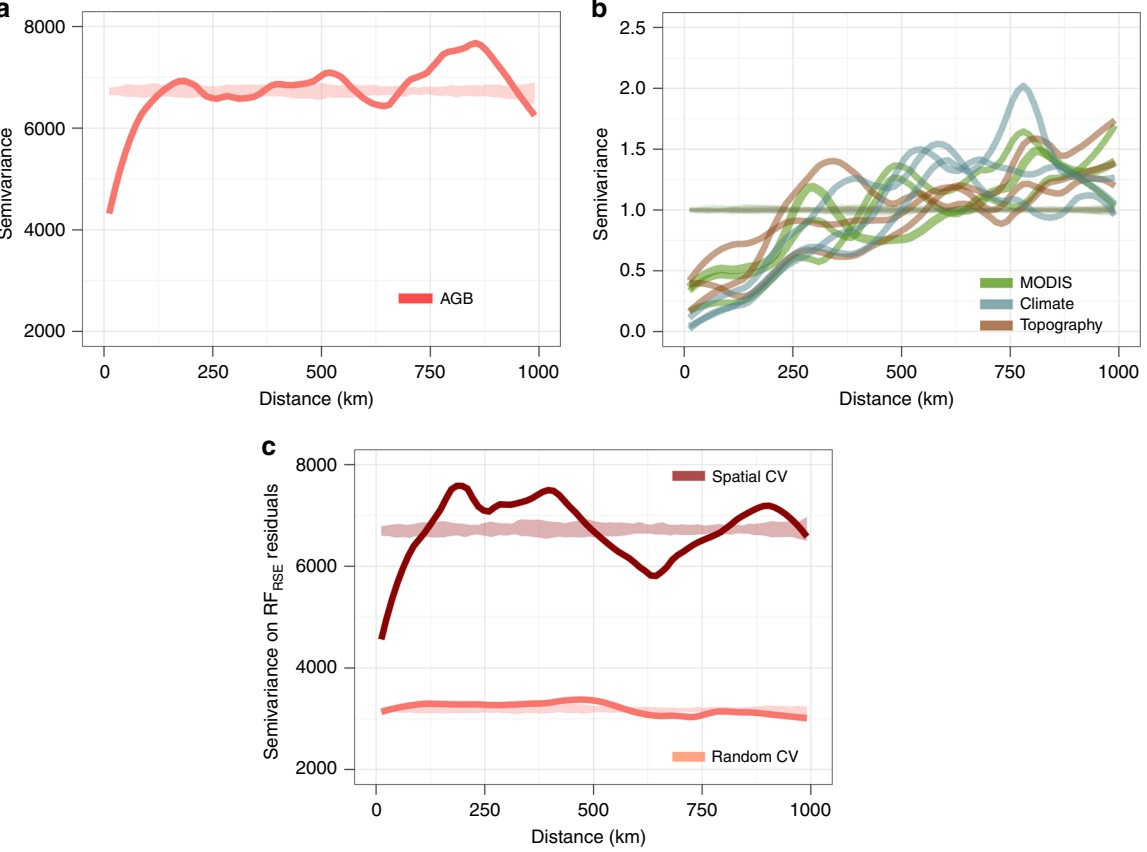

**Fig. 2 Spatial autocorrelation within model input variables and residuals.** Semivariograms showing spatial autocorrelation of reference pixels AGB (**a**), AGB predictors (**b**), and AGB model residuals (**c**). In **b**, we split predictors into three groups (i.e., MODIS, climate, and topography) and only displayed three predictors by group for illustration purposes. In **c**, we computed AGB model residuals using a random 10-fold cross-validation (light red) and a spatial 44-fold cross-validation (dark red).

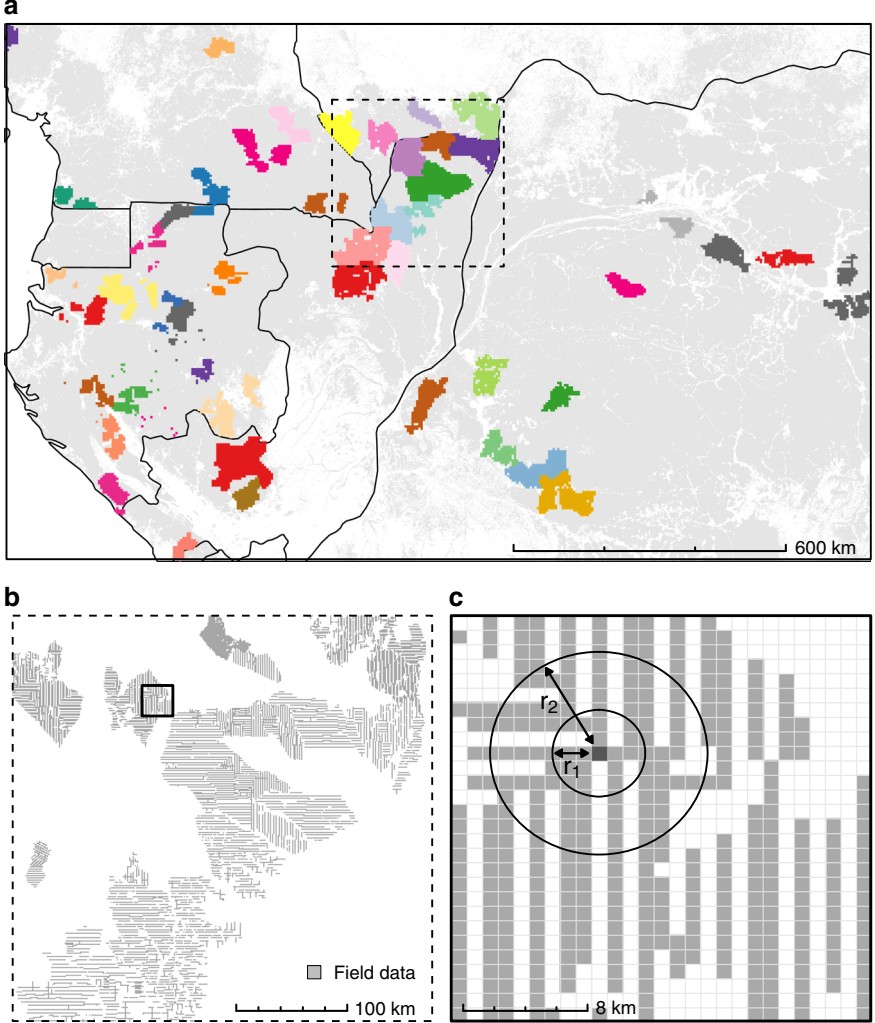

**Fig. 3 Assessment of predictive performance in a spatially structured environment. a** Clustering of field data into 44 spatial folds (bright colors) used in spatial $K$-fold CV, superimposed over the spatial distribution of moist forests[61] (light gray) and country borders. **b** At the regional scale (a few hundred km, clipped from **a**), field data (light gray) are aggregated into dense clusters, leaving large swaths of unsampled areas. The outlined region is further expanded in **c**. **c** At the landscape scale (a few tens of km), reference pixel AGB and AGB covariate values are spatially dependent (as seen in Fig. 2a, b), which violates the independence hypothesis of training and test sets in cross-validation procedures based on random data splitting. We used circular buffers of increasing radii $r$ to exclude training data (gray) located around the test data (dark gray) in a leave-one-out CV (B-LOO CV) to evaluate how spatial dependence in the data impacted CV statistics.

We thus investigated the consequences of ignoring spatial dependence in the data when evaluating the predictive power of AGB mapping models by contrasting the results of cross-validation procedures ignoring (random $K$-fold CV) or accounting for spatial autocorrelation. Following recent methodological guidelines[30,31], we tested two alternative approaches (see "Methods" section for details) that stem from the fairly simple objective to increase the spatial distance and therefore independence between training and test sets of data compared to a classical random $K$-fold CV. The first approach consists of splitting observations into $K$ sets based on their geographical locations rather than at random to create spatially homogenous clusters of observations. Spatial clusters (Fig. 3a) are then used $K$ times alternatively as training and test sets for cross-validation (hereafter spatial $K$-fold CV). The second approach is similar to a leave-one-out cross-validation (i.e., where the test set consists of a single observation) but includes spatial buffers around the test observations (hereafter B-LOO CV)[32]. Spatial buffers are used to remove training observations in a neighboring circle of increasing radii around the test observations (illustrated in Fig. 3c), thereby

assuring a minimum (and controlled) spatial distance between the two sets.

Our results show that ignoring spatial dependence in the data conceals poor predictive performance of the mapping model beyond the range of autocorrelation in forest AGB, leading to false confidence in the resulting map and erroneous assessment of predictor importance, contributing to the debate over the utility of MODIS data in this context[23,33]. This very general methodological issue is also discussed in the broader context of global mapping models with "Big Data" approaches recently at the forefront of ecology literature.

## Results

**Ignoring data spatial structure in cross-validation**. We built an RF model to predict AGB from a combination of nine MODIS and 27 environmental variables ($RF_{RSE}$, see the "Methods" section for a list of variables) and assessed model predictive performance with a classical random 10-fold CV procedure, i.e., ignoring any spatial dependence structure in the data. This led to an estimated $R^2$ of 0.53 and a mean prediction error (RMSPE) of 56.5 Mg ha$^{-1}$

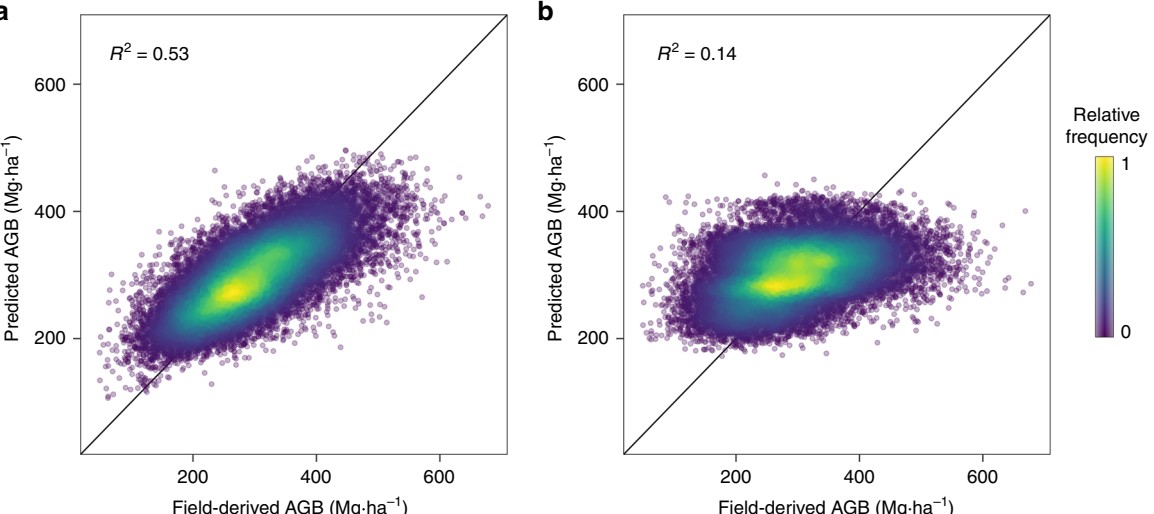

**Fig. 4 K-fold CV of AGB predictions based on MODIS and environmental variables (RF$_{RSE}$).** Heat plots showing the relationship between observed AGB versus RF$_{RSE}$ AGB predictions from (**a**) random 10-fold CV and (**b**) spatial 44-fold CV.

(19%). Model predictions showed a fairly linear relationship with observed AGB, although the model tended to overestimate low AGB values and underestimate high AGB values (Fig. 4a), a common bias pattern with the RF algorithm for which bias-correction methods have been proposed[34]. One could, therefore, conclude from the random K-fold CV that dense tropical forest AGB can indeed be predicted at 1-km resolution from MODIS and environmental data while constraining the prediction error to <20% on average.

**Accounting for data spatial structure in cross-validation.** We then tested the same RF model with a spatial K-fold CV approach to assess the influence of spatial autocorrelation in the data on the statistics of the model predictive power. In this approach, reference AGB pixels were split into 44 homogeneous spatial clusters (Fig. 3a) based on a maximum distance threshold of 150 km for observations within clusters (i.e., a slightly longer distance than the range of autocorrelation of forest AGB; Fig. 2a), and clusters were alternatively used as training and test sets. The spatial 44-fold CV led to a sharp decline in model $R^2$ ($R^2 = 0.14$, Fig. 4b) and an increase in RMSPE (i.e., 77.5 Mg ha$^{-1}$, or 26%). By comparison, a null model that systematically predicted the mean of the training observations regardless of test observation covariate values (i.e., a model with no predictors but an intercept) yielded an RMSPE of 82 Mg ha$^{-1}$ (27.7%). A comparison of model residuals derived from random and spatial K-fold CVs also provides important insights into the insidious effect of spatial autocorrelation[30]. Indeed, a common model diagnosis practice consists of looking for spatial autocorrelation in model residuals. If present, a range of statistical methods can be applied to minimize deleterious effects on model parameter estimation[35]. Here, even after a random 10-fold CV, the residual structure was completely absorbed in RF$_{RSE}$'s predictions (Fig. 2c). Diagnosis methods that target model residuals would thus fail to detect any problem in the model's specification and lead to undue confidence in model predictions.

While implementing a spatial K-fold CV is straightforward, it requires choosing a size for spatial clusters, which defines the average distance between training and test data sets. A more insightful approach, though computationally demanding, consists of assessing the model predictive power for a range of increasing distances between training and test data. In this buffered

leave-one-out cross-validation (B-LOO CV) approach[32], model validation is performed on one test observation at a time, which is buffered by omitting from the training set neighboring observations in circles of increasing radii (Fig. 2e). We thus implemented a B-LOO CV on 1,000 randomly selected test observations, considering each time a buffer distance from 0 to 150 km. In the absence of an exclusion buffer, i.e., for a buffer radius of 0 km between training and test observations, the B-LOO CV led to an $R^2$ of 0.50 on average, in good concordance with results obtained with a random 10-fold CV. However, increasing the size of the exclusion radius - hence weakening the spatial dependence between training and test observations - led to a rapid decrease in model $R^2$ (Fig. 5a) and a concomitant increase in model RMSPE (Fig. 5b). For test pixels located at c. 50 km of the nearest training pixel, $R^2$ decreased to c. 0.15 on average - close to what was found with the spatial K-fold CV - and became virtually null beyond 100 km. First and foremost, this result shows that there is a strong spatial component in the variance of pixel AGB explained by the RF$_{RSE}$ model. This spatial component cannot be predicted by the model so that its predictive performance actually decreases with decreasing spatial autocorrelation between training and test data (as illustrated in Fig. 4c). Second, this result suggests that the cluster size set in the spatial K-fold CV was not sufficiently large to move all test pixels out of the autocorrelation range of training pixels.

A further important question is to quantify the share of model performance actually pertaining to the considered predictors. We compared RF$_{RSE}$ to a purely spatial model taking pixel geographic coordinates as sole predictors (RF$_{XY}$). RF$_{RSE}$ and RF$_{XY}$ showed very similar predictive performance, both in terms of absolute $R^2$ and RMSPE values, which similarly varied with buffer radii (Fig. 5a, b). RMSPE reached a plateau for pixels located beyond 100 km of the nearest training pixel, which coincided with the average prediction error of the null model. Taken together, these results show that the MODIS and environmental variables used in RF$_{RSE}$ have weak, if any, predictive power over forest AGB variation at the 1-km scale. Rather, RF$_{RSE}$ likely leans on the geographical proximity and the resulting correlation between training and test pixels to exploit the spatial distributions of the model's predictors to predict the pixel AGB. Simply put, the model predicts a given pixel AGB based on its spatial - not optical and environmental - proximity to training pixels. An intuitive illustration of this phenomenon can be obtained by taking the

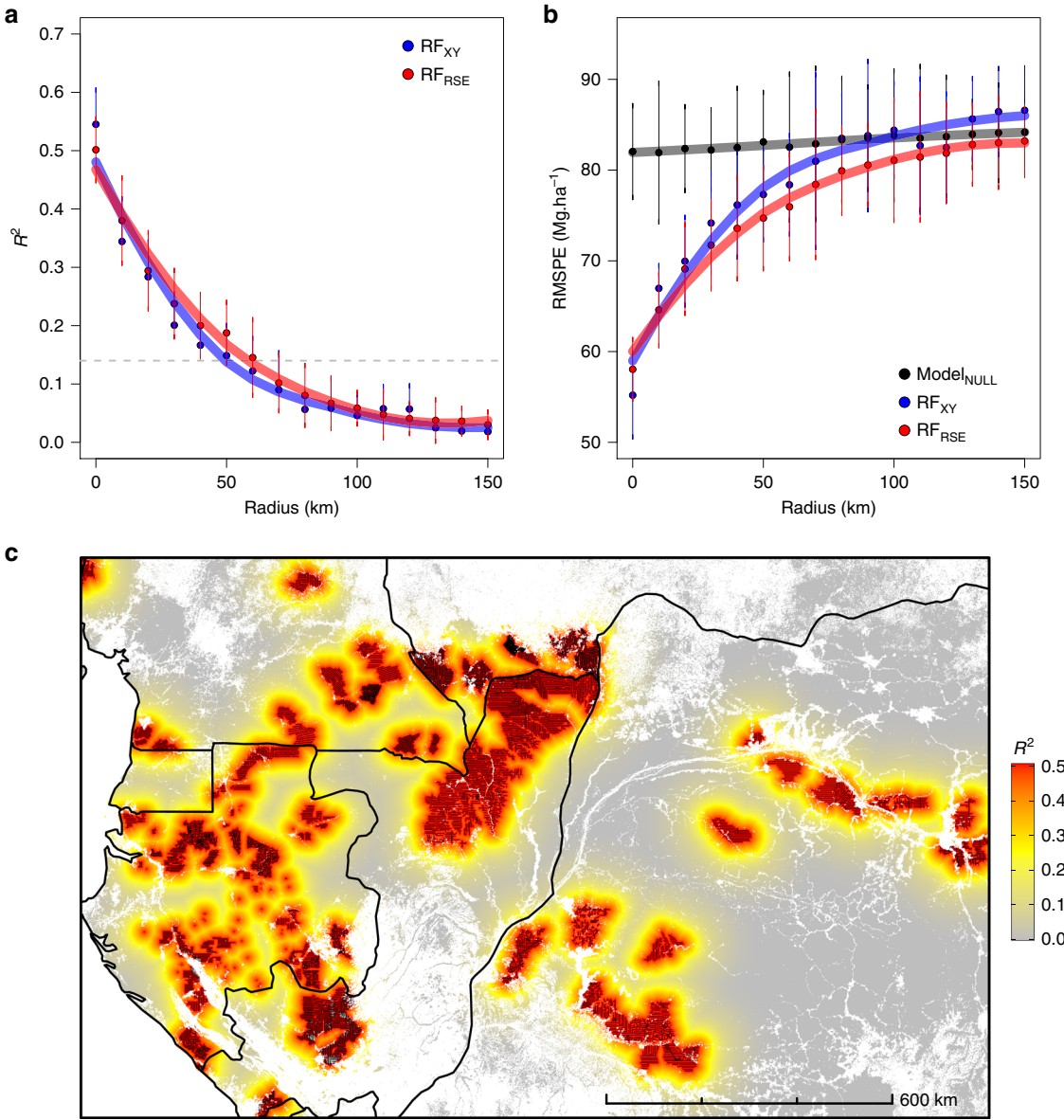

**Fig. 5 Influence of data spatial structure on the model's CV statistics. a** Change in the coefficient of determination (mean $R^2 \pm$ SD over $n = 10$ iterations, see "Methods" section for details) between predicted and observed pixel AGB as buffer radii for neighboring pixel exclusion increases in the buffered leave-one-out cross-validation (B-LOO CV). Predictions are made with the model based on MODIS and environmental variables (RF$_{RSE}$, red) and the model based on the pixels' geographic coordinates (RF$_{XY}$, blue). The gray dashed line represents the $R^2$ obtained by spatial $K$-fold CV in Fig. 4b. **b** Change in the root mean squared prediction error (mean RMSPE $\pm$ SD over $n = 10$ iterations, see "Methods" section for details) in the B-LOO CV. In addition to RF$_{RSE}$ and RF$_{XY}$, the RMSPE of a null model that systematically predicts the mean of training data is plotted (Model$_{NULL}$, black). **c** Projection of RF$_{RSE}$ prediction $R^2$ in the study area. The distance between each map pixel to the closest reference AGB pixel (represented in black) is converted to a mean $R^2$ using the relationship displayed in **a**.

reverse approach of predicting geographical coordinates using the optical and environmental predictors. Doing this exercise, we obtained a RMSE on the predicted coordinates of about 25 km. This means that the combinations of predictor variables are sufficiently unique to predict pixel approximate location, and thereby predict the AGB.

## Discussion

In this study, we used the largest known data set of reference AGB estimations derived from field data in tropical forests and widely used environmental and optical RS data to undertake spatial prediction using a common machine-learning (ML) approach. Our results illustrate a serious methodological flaw in a number of large-scale ecological mapping studies, where the predictive power of ML models is evaluated using nonspatial cross-validation.

Assessing model predictive power is indeed a key step in ecological mapping studies, as the purpose of such models is precisely to extrapolate the variable of interest beyond sampling locations. A classical approach to evaluate whether predictive models are transferable is to withhold a random subset of observations for model testing (e.g., a random $K$-fold CV). This approach has been common practice for a decade in pantropical AGB mapping studies[2,3] and is broadly used in the prolific field of Big Ecology[36]. This approach, which assimilates test observations to independent new observations, is inherently invalid in a geospatial context because ecological data are almost always autocorrelated[37]. This means that observations located within the

range of spatial autocorrelation in the data, which exceeds 100 km in our case study on forest AGB in central Africa, should be treated as pseudoreplicates rather than new observations in model validation schemes. Ignoring spatial dependence in model cross-validation can create false confidence in model predictions and hide model overfitting, a problem that has been well documented in the recent literature on statistical ecology[30,32,35,38]. This issue was particularly evident in our results: (1) the model explained more than half of the AGB variation in the vicinity of sampling locations but completely failed to predict AGB at remote locations, showing that it was strongly locally overfitted, and (2) the random K-fold CV scheme did not allow the diagnosis of model overfitting and provided overly optimistic statistics of model predictive power. It is even likely that ML algorithms that optimize the local adjustment of predictions in the variable space[39] amplify the problem. Indeed, overfitting was less obvious when using linear modeling approaches with the same data set (e.g., Supplementary Fig. 1).

The methodological flaw documented here likely affected previous pantropical AGB mapping studies based on ML algorithms[2,3], which may largely explain their disagreements. Indeed, reference sets of AGB observations in these studies were derived from space-borne LiDAR data from the Geoscience Laser Altimeter System (GLAS), assuming these data provide, when calibrated with field measurements, accurate estimates of forest AGB[40]. The GLAS sampling layout takes the form of a set of footprints regularly spaced out every 170 m along tracks but separated by tens of km across tracks. It is, therefore, reasonable to think that once aggregated at the mapping resolution (i.e., 500 m or 1 km), the distance between observed AGB pixels along GLAS tracks is considerably smaller than the range of autocorrelation in the data (as illustrated in Supplementary Fig. 2). This layout likely facilitates overfitting by ML algorithms, in a way similar to what was observed in our study case. Proving this claim is, however, not possible, as the data used in these studies, including the geolocation of reference AGB data, which have been heavily filtered through quality-check procedures, have not been publicly released. However, our results show that including high-quality annual composite MODIS images as predictors in $RF_{RSE}$ did not permit the establishment of a predictive relationship for closed-canopy, dense tropical forest AGB, despite a fivefold variation factor in the reference data set (c. 100–500 $Mg\,ha^{-1}$). Our observations shed strong doubts on the pertinence of such variables to spatialize AGB variations in closed-canopy tropical forests[3], let alone to monitor subtle changes in carbon dynamics associated with degradation or growth[24]. Rather, our results likely reflect the well-known saturation of medium- and high-resolution passive optical data[23,41] beyond 100 $Mg\,ha^{-1}$ and confirm the overoptimistic assessment of MODIS-based AGB models made in previous studies - an issue that has already been raised in the very thematic of large-scale forest AGB mapping[42].

It should, however, be emphasized that our results are contingent upon the MODIS variables used in this study, which, for instance, do not account for intra-annual variation in vegetation reflectance (as in, e.g., ref. [2]), and we cannot exclude that other approaches to exploit the MODIS signal may improve its potential for forest AGB inference. We also do not claim that the AGB maps published in[2,3] are necessarily wrong: they would simply not be better than the spatial kriging of the GLAS data because of the low predictive power of underlying mapping models beyond the range of data autocorrelation. This suggests that projections between GLAS tracks are likely very uncertain and that models are not transferable to unknown locations, whether in space (i.e., to a different region) or in time (i.e., to future conditions, e.g., based on a change in MODIS reflectance).

Going forward, recent and upcoming Earth observation missions dedicated to documenting variations in forest structure by space-borne LiDAR (the Global Ecosystem Dynamics Investigation) or SAR (BIOMASS missions) will undoubtedly greatly advance our understanding of forest AGB distribution across the tropics. Sampling of the territory by airborne or UAV-LiDAR scanning is also becoming denser across the tropics[43–47]. At the same time, this study shows that a better valorization of commercial forest inventory data is possible, while concerted efforts are ongoing to develop national forest inventories and improve the coordination between scientific plot networks[48,49]. We argue that the unprecedented amount of data produced by these efforts will provide a unique opportunity to clarify, develop, and extensively validate a conceptual modeling framework for biomass mapping from high- or medium-resolution multispectral data, which could allow leveraging decades of image archives.

Beyond the context of forest AGB mapping, our analysis clearly shows that a random K-fold CV does not provide sufficient information to demonstrate the predictive ability of a mapping model. This questions recent high-impact publications of several global-scale maps[50–52] that completely overlooked the issue of data autocorrelation. A recent example is a global-scale map of nematode worms[50], which was built with an RF model based on a set of 73 environmental, optical, and anthropogenic predictors. Following a random 10-fold CV, the authors obtained an $R^2$ of 0.86, which they interpreted as a sign of high model predictive strength. Using their publicly available data set and a purely spatial RF model (based on geographic coordinates alone), we obtained fairly similar results ($R^2 = 0.8$, Supplementary Fig. 3), indicating that simple spatial kriging would be as predictive as the proposed model and that any interpretation of the ecological determinism of these organisms should be done with extreme caution. We thus would like to draw attention to the fact that in an ecological or biological context, where processes inherently create a positive or negative correlation between neighboring locations, rigorous, spatially explicit assessment of ML models is required and should become the norm for every large-scale mapping study.

## Methods

**Reference aboveground forest biomass data**. The reference set of forest aboveground biomass (AGB) pixels, i.e., The Congo basin forest AGB (CoFor-AGB) data set has recently been publicly released[18]. CoFor-AGB was built from management forest inventories of 113 logging concessions spread across dense forests of central Africa (Fig. 1). In total, CoFor-AGB contains 191,563 plots, covering a cumulative sampling area of 94,513 ha and representing $11.82 \times 10^6$ tree measurements and 1,091 identified taxa. A standardized computation scheme was applied across all inventories to estimate plot AGB for all trees ≥10 cm in diameter at breast height. The computation scheme was evaluated using higher-quality scientific inventory data[18]. The estimated average error of the 0.5-ha management forest plot AGB was 15%, which was only moderately higher than that based on scientific inventory plots (c. 8.3%)[18]. Plot AGB estimations were aggregated at the 1-km spatial resolution, resulting in 59,857 unique pixels across the study area.

**Environmental and MODIS data**. To create spatial predictive models of forest AGB, we stacked freely available and seemingly relevant global map layers characterizing local environmental conditions (including climate, topography, and soil types) and MODIS-derived vegetation reflectance properties. Covariates were cropped to the extent of the study area (Fig. 1), aligned and resampled to a 1-km spatial resolution.

Concerning climate conditions, we first assembled monthly average statistics of precipitation (P), temperature (T), solar radiation (SR), and water vapor pressure (WP) from the Worldclim2 database[27] for the period 1970–2000, monthly potential evapotranspiration (PET) from the Global-PET database[53] and statistics of annual cloud cover (CC) frequency derived from MODIS[54]. Monthly P and T were used to compute the 19 standard WorldClim bioclimatic variables, including annual trends (i.e., the mean of monthly means), seasonality (i.e., standard deviation and coefficient of variation of monthly means for T and P, respectively) and indicators of extreme conditions (e.g., mean temperature of the driest quarter). Annual trends and seasonality (i.e., standard deviation of monthly means) were also computed for SR and WP. We further computed several climate layers from monthly P and PET, including an index of annual water availability (WA, defined as the ratio of cumulative P to PET)[55], the maximal climate water deficit[56], and a

**Table 1 Mean and range of forest AGB and AGB covariates.**

| Code | Description | Mean (range) |
|---|---|---|
| 1. AGB | Aboveground biomass (Mg ha$^{-1}$) | 296 (49–678) |
| 2. T_am | Annual mean temperature (°C) | 24.3 (22.1–26.5) |
| 2. Prec_am | Annual mean precipitation (mm) | 1713.6 (1125.9–2864.5) |
| 2. CC_am | Annual mean cloud cover frequency (%*100) | 7806.5 (6246.4–9758.3) |
| 2. Vapor_m | Annual mean water vapor pressure (kPa) | 2.5 (2.2–2.9) |
| 2. SolRad_m | Annual mean solar radiation (kJ m$^{-2}$ day$^{-1}$) | 14,705.8 (11,500–18,236.2) |
| 2. WA | Water availability (unitless) | 1 (0.8–2) |
| 3. T_seaso | Temperature seasonality (standard deviation*100) (°C) | 78.7 (40.7–185.9) |
| 3. Prec_seaso | Precipitation seasonality (coefficient of variation) (mm) | 51.8 (22.4–83) |
| 3. Vapor_sd | Standard deviation of monthly water vapor pressure (kPa) | 0.1 (0–0.3) |
| 3. SolRad_sd | Standard deviation of monthly solar radiation (kJ m$^{-2}$ day$^{-1}$) | 1410 (546.7–2251.2) |
| 4. T_mdq | Mean temperature of the driest quarter (°C) | 23.6 (20.3–26.2) |
| 4. T_mwarmq | Mean temperature of the warmest quarter (°C) | 25.2 (23.2–27.5) |
| 4. Prec_dm | Precipitation of the driest month (mm) | 31.2 (0–105.6) |
| 4. DS_long_length | Long dry season length (number of months) | 4.4 (2–6) |
| 4. DS_cumu_length | Dry season(s) length (number of months) | 5 (2–8) |
| 4. DS_long_sever | Long dry season severity (mm) | 316.8 (49.1–602.6) |
| 4. DS_cumu_sever | Dry season(s) severity (mm) | 332.6 (49.1–602.6) |
| 4. CWD | Maximal climate water deficit (mm) | −172.7 (−558.8–0) |
| 4. Vapor_DS_m | Mean monthly water vapor pressure during the longest dry season (kPa) | 2.4 (2–3) |
| 4. Vapor_DS_max | Max. monthly water vapor pressure during the longest dry season (kPa) | 2.5 (2.2–3) |
| 4. SolRad_DS_m | Mean monthly solar radiation during the longest dry season (kJ m$^{-2}$ day$^{-1}$) | 14613.2 (8971.9–19218.7) |
| 4. SolRad_DS_max | Max. monthly solar radiation during the longest dry season (kJ m$^{-2}$ day$^{-1}$) | 15519.6 (9572.9–20211.4) |
| 5. Elev | Elevation (m) | 532.6 (102–935.4) |
| 5. Slope | Slope (°) | 9.3 (1–43.2) |
| 5. HAND | Height above the nearest drainage (m) | 35.4 (0–449.9) |
| 5. Convexity | Convexity (unitless) | 39.9 (9.8–53.6) |
| 6. Soil_types | Soil types (unitless) | – |
| 7. R_mean | RED (mean - unitless) | 0.048 (0.000–0.254) |
| 7. NIR_mean | NIR (mean - unitless) | 0.444 (0.291–0.567) |
| 7. G_mean | Green (mean - unitless) | 0.067 (0.000–0.158) |
| 7. SWIR1_mean | SWIR 1 (mean - unitless) | 0.44 (0.288–0.534) |
| 7. SWIR2_mean | SWIR 2 (mean - unitless) | 0.234 (0.156–0.306) |
| 7. SWIR3_mean | SWIR 3 (mean - unitless) | 0.086 (0.000–0.196) |
| 7. SWIR3_sd | SWIR 3 (sd - unitless) | 0.03 (0.004–1.14) |
| 7. EVI2 | EVI2 (unitless) | 0.635 (0.238–0.857) |
| 7. NDII | NDII (unitless) | 0.676 (0.308–1.032) |

Variables are split into seven categories, namely, the dependent variable (1) and covariates representing mean climate conditions (2), climate seasonality (3), extreme climate conditions (4), topography (5), soil types (6), and MODIS reflectance (7).
Topographic variables were computed using 30-m STRM data over 1-km$^2$ pixels.

simple metric of monthly water deficit (defined as P PET). The monthly water deficit metric was used to identify dry season months (with a negative deficit) and to compute dry season-specific climate layers, including the length (number of months) and severity (cumulative water deficit, mean and maximum SR and WP) of either all dry season months throughout the year or only months pertaining to the longest dry season. The full set of climate variables was then submitted to principal component analysis to identify and remove highly collinear variables along the principal climate variation axes. We finally retained the 22 variables listed in Table 1. Topographic variables were computed using the 30 m SRTM data and included the average elevation, slope, height above the nearest drainage (HAND)[57], and convexity within the 1-km pixels. Soil types were extracted from the Harmonized World Soil Database (HWSD)[58] and merged into a class containing Arenic Acrisols (sandy soils) and a mixed class containing all other soil types following ongoing analyses on the influence of soil types on the spatial distribution of forest floristic and functional compositions across the Congo basin.

Vegetation reflectance data corresponded to the MODIS Collection 6 product processed using the latest version of the MAIAC algorithm[26]. The MODIS data cube was generated at a 1-km resolution as a stack of 8-day composite images for the 2000–2010 period, which corresponds to the period of field data collection. For each pixel, we retained only the best quality observations based on aerosol optical depth (AOD) and bidirectional reflectance function (BRF) quality flags and generated 10-year reflectance mean and standard deviation layers for each spectral band. Following[3], we used the reflectance mean layer to compute EVI2 and NDII vegetation indices and selected the same 9 reflectance variables as AGB covariates (Table 1).

**Spatial forest biomass modeling and pixel exclusion.** The set of reference AGB pixels was filtered to focus the analysis on forested pixels with stable canopy cover from 2000–2010 and reliable pixel-level AGB estimations. First, we used the 30-m Landsat forest classification for the year 2000[5] to remove pixels with less than 89% vegetation cover at the beginning of the study period (i.e., 11,566 pixels or c. 19.3% of the initial data set) and hence reduce the potential indirect influence of canopy cover on the relationship between reflectance variables and forest AGB[23]. The choice of the 89% threshold was a trade-off between maximizing pixel forest cover and avoiding discarding all pixels on the western side of the study area, where all

forested lands are assigned a 90% vegetation cover in the Landsat product, which likely reflects atmospheric pollution in optical Landsat data. Second, to maximize the representativity of field plots within the pixels, we removed all pixels containing fewer than 3 field plots (i.e., 19,240 pixels or 39.8% of the filtered data set). This resulted in a minimal sampling rate of 3 * c. 0.5 ha by 1 km pixels (with a mean ± standard deviation of 4.14 ± 1.03 plots), equivalent to the rate found in pantropical AGB mapping studies (e.g., 5 * c. 0.25 ha GLAS shots[2]). Third, acknowledging that any substantial change in pixel land cover during the period covered by the MODIS image composite (2000–2010) might unduly blur the relationship between field-derived AGB and MODIS reflectance variables (e.g., conversion of forests to cultures after the time of field inventories), we also excluded pixels characterized by (i) a cumulative forest loss between 2000 and 2010 greater than 5% of the pixel area and (ii) a nonnull percentage of fire occurrence between 2000 and 2012 (736 pixels or 2.5% of the filtered data set). Forest loss data were obtained from the 30-m Landsat vegetation cover change product[5]. The percentage of fire occurrence data was obtained from the burnt area product (version 2) of the Land Cover project of the European Space Agency Climate Change Initiative (ESA CCI-LC)[59]. Finally, we screened pixel spectral responses in each MODIS variable and removed obvious outliers, defined as pixels whose spectral value was more than 10 standard deviations from the mean (90 pixels or 0.2% of the filtered data set). The final data set contained a total of 28,225 unique pixels (Supplementary Fig. 4) that were used as input in Random Forest[25] for spatial forest AGB modeling.

**Assessing model prediction error.** We used three strategies to cross-validate models and generate statistics of the model predictive power (Fig. 6).

The first strategy corresponded to a common K-fold cross-validation whereby observations were randomly split into K sets (random K-fold CV), ignoring any structure of spatial dependence in the data. Model training was then performed iteratively on K-1 sets, each time withholding a different set for testing. The vector of so-called independent AGB predictions was then used to generate CV statistics, namely, the squared Pearson's correlation between observed AGB values and AGB predictions (noted $R^2$) and the root mean squared prediction error (RMSPE). Here, we used $K = 10$, a common choice made by modelers.

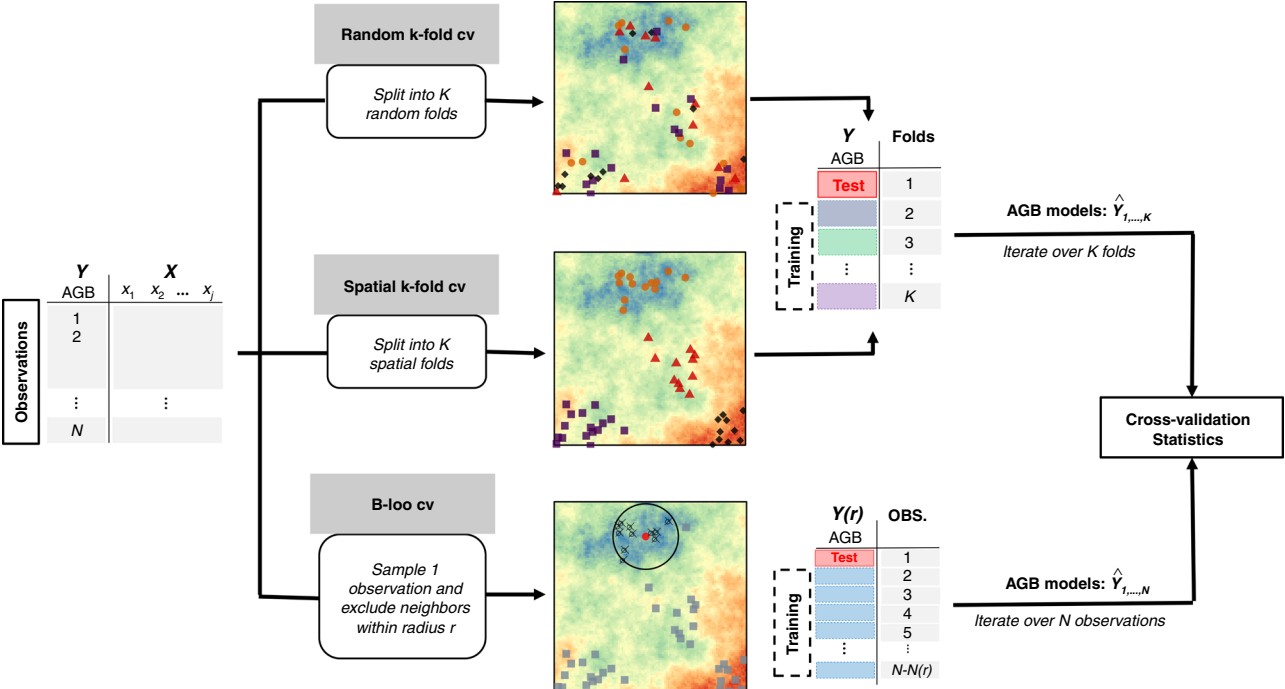

**Fig. 6 Workflow of model cross-validation strategies.** Schematic illustration of the three strategies used for model cross-validation (random *K*-fold CV, spatial *K*-fold CV, buffered leave-one-out CV). In *K*-fold CVs, observations within the same fold are represented with a similar symbol and color. In the B-LOO CV, the test observation is represented as a red dot, training observations as grey squares, and observations within the exclusion buffer (black circle) are crossed.

The second strategy, i.e., spatial *K*-fold CV, differs from the random *K*-fold CV in the way observations are split into spatially structured sets. Here, the objective is to group observations into spatially homogeneous clusters of larger size than the range of autocorrelation in the data to achieve independence between CV folds. Spatial clusters were generated using a hierarchical cluster analysis (complete linkage method) of the distance matrix of pixel geographical coordinates and a clustering height (i.e., the maximum distance between pixels within each cluster) of $H = 150$ km, i.e., a slightly longer distance than the range of autocorrelation of forest AGB (Fig. 2a).

The third strategy, i.e., the buffered leave-one-out cross-validation (B-LOO CV), is inspired by the leave-one-out cross-validation scheme, in that a single observation is withheld for model testing per model run. In the case of B-LOO CV, however, observations within a distance $r$ from the test observation are excluded from the model training set. Training and testing the model for a range of $r$ values allows investigation of the influence of spatial proximity between test and training observations on model prediction error. Here, we considered 16 $r$ values (from 0 to 150 km by 10 km); hence, the model was calibrated and tested 16 times per test observation. To generate B-LOO CV statistics presented in Fig. 5a, b, we (1) generated AGB predictions for 100 randomly selected test observations (i.e., 1600 model runs), allowing computation of the model's $R^2$ and RMSPE for each $r$ value, and (2) repeated this procedure ten times (i.e., 16,000 model runs) to provide the average and standard deviation of CV statistics over the 10 repetitions. It is worth noting that we integrated a safeguard against predictive extrapolation within the iterative B-LOO CV procedure. Because the geographical and environmental spaces are closely linked, removing training observations in the spatial neighborhood of a test observation may remove the environmental (and optical) conditions found at that test location from the model's calibration domain. The model's prediction at that test location would thus amount to a case of predictive extrapolation (i.e., forcing the model to predict outside the calibration domain), leading to an inflation of model error that we did not intend to consider here. For each randomly selected test observation, we thus first removed neighboring observations at the largest $r$ value and verified that optical and environmental conditions at the test location still fell within the range of values found in the model's calibration domain. If not, we discarded the test observation and randomly selected a new observation.

**Predicting pixel geographic location**. We used a multivariate implementation of the random forest algorithm[60] to predict pixel geographic coordinates (i.e., latitude and longitude) from the set of optical and environmental predictors listed in Table 1. The model was trained on 500 pixels randomly drawn from the full set of reference AGB pixels.

**Reporting summary**. Further information on research design is available in the Nature Research Reporting Summary linked to this article.

## Data availability

All data analyzed in this study are publicly available. The raster of observed AGB pixels is available through figshare (https://doi.org/10.6084/m9.figshare.11865450). Environmental data used in this study were obtained from the following sources: Worldclim (https://worldclim.org/data/worldclim21.html), Global-PET (https://cgiarcsi. community/data/global-aridity-and-pet-database/), Cloud Cover (http://www.earthenv. org/cloud), SRTM (http://srtm.csi.cgiar.org/srtmdata/), and Harmonized World Soil Database (HWSD, http://www.fao.org/soils-portal/soil-survey/soil-maps-and-databases). The MODIS MAIAC product cropped over the study area is available through figshare (https://doi.org/10.6084/m9.figshare.12751628). Data sets used to filter pixels were obtained from the following sources: Global Forest Change (https://data. globalforestwatch.org/datasets/tree-cover-2000) and CCI-LC database (Burn Area product v.2.0, 2000–2012 epoch, http://maps.elie.ucl.ac.be/CCI/viewer/download.php). An access link to the nematode worm data set is provided in the original publication (https://gitlab.ethz.ch/devinrouth/crowther_lab_nematodes). The GLAS data used in Supplementary Fig. 2 are available from www.theia-land.fr/en/product/lidar/.

## Code availability

The analysis scripts for the random cross-validation, spatial cross-validation, and buffered leave-one-out cross-validation are available through figshare (https://doi.org/10.6084/m9.figshare.12790085).

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

## Acknowledgements

P.P. was supported by a postdoctoral grant from the 3DForMod project. 3DForMod was funded in the framework of the ERA-NET FACCE ERA-GAS (ANR-17-EGAS-0002-01), which has received funding from the European Union's Horizon 2020 research and innovation program under Grant Agreement No. 696356.

## Author contributions

P.P., F.M., and R.P. conceived the study, analyzed the data, and led the writing of the paper. S.G.F., N. Bay., and G.C. collated field data. A.L. and G.V. processed MAIAC data. M.R.-M., N. Bar., N.P., V.R., and C.D. commented on the analyses and provided critical inputs for the writing of the paper.

## Competing interests

The authors declare no competing interests.
