## [Peer Review File · Nature Communications]

Reviewers' Comments:

Reviewer #2:

Remarks to the Author:

This is an excellent, important, and much-needed study that reveals central problems with commonly used algorithms for mapping forest biomass and other ecosystem variables. Essentially, it shows that the apparent high predictive power of these algorithms is due to the way cross-validation is conducted, with validation points close in space to training points, to spatial autocorrelation in the response variable, and the fact that combinations of the predictor variables can be used as proxies for location. Many high-profile results need to be reconsidered in light of these findings. My only concerns are fairly minor, and are in regard to some areas where I believe the methods and/or results need to be better explained to the reader.

As the authors surely know, spatial structure in independent variables and response variables alone (of the kind shown in figures 2a,b) is not necessarily a problem for most statistical analyses – what is a problem is spatial autocorrelation in residuals. The graphs in figure 2a,b and associated text are potentially misleading in this sense, as they focus on spatial autocorrelation in the variables. Figure 3c shows the spatial autocorrelation in the residuals, which is the real problem, and also that these appear uncorrelated in the typical (problematic) cross-validation approaches. The text explaining this figure and its significance could be clearer; these are important points to get across and deserve some more explanation (lines ~190-200)

I struggled to figure out exactly what was being done in some of the analyses, and was surprised not to be able to find some details even in the SI. In particular, I could find nothing in the main text to explain how figure 4c is constructed, and it was hard for me even to figure out the explanation in the SI. It also took me a while to understand the results shown in figures 4a,b. (At first I thought that the cross-validation was as usual being done on a random 10% of the data, and that the analyses were leaving out ever more data around that 10% from the testing data, so I wondered if the result was an artefact of having less and less testing data to work from. Then finally I reached the text late in SI explaining that only one point was evaluated at a time, but only after I'd been wondering about this for a while...)

In terms of interpreting results, how do you explain that the r^2 values in figure 4a drop even below those from the spatial cross-validation? Does this mean that the performance of the spatial cross-validation is itself still inflated by spatial autocorrelation across the borders of the patches that are included or dropped?

The authors note that overfitting is not as big a problem when a generalized linear model is used (265-267). Personally, I would be very interested to see that parallel result. I hope the authors intend to publish that somewhere, even if not here.

Additional specific comments

Figure 2. It might be better to split this into 2 or more separate figures. In particular, it would be nice to see one figure devoted exclusively to the semivariograms, and to have that figure better show the pattern at relatively short spatial scale. For example, in figure 2a, the scale is such that it is hard to tell at what distance the semivariance of AGB (dark red line) asymptotes. Many empirical ecologists will be interested to know the shape of the relationships in the first 250 km in greater detail.

Figure 2c requires a distance scale.

Lines 238-239. It would be interesting to know a bit more about the fitted RF models. Is there any way the general issues could be diagnosed from the structure of the RF models? For example, if the models in part take the form of a decision tree, could one map out the branches of the decision tree and see that the combinations are essentially proxies for spatial location?

Lines 254-55. Saying all points within 100 km are pseudoreplications seems overstated... the key point is not the spatial autocorrelation of the independent or dependent variables, but of the residuals. That seems to be somewhat shorter, isn't it? It would be nice to see figures 2a,b, and figure 3c in the same figure, and to see the first part (<200 km) of each magnified to better see at what distance the semivariance asymptotes in each case.

Table 1. What makes the mean temperature annual? That is, isn't it just mean temperature over some time period? Same for other "annual" variables. For seasonality, need to specify somewhere the time scale over which standard deviations and CVs of temperature and precipitation are calculated. Often, for example, it is the standard deviation among months of monthly mean (over years) temperatures. MODIS reflectance – specify units. Topography – specify the spatial grain of the slope, convexity, etc. That is, the same point can have a different slope if it is the slope at 1 km scale vs. 100 m vs. 10 m. And similarly, a point can be on a small ridge (10 m scale) within a valley (1 km scale), so need a spatial grain for convexity.

421. Would it make sense to repeat this for different cluster sizes?

440-445. Great safeguard!

449-451. I recommend making the code used for this analysis publicly available at time of publication.

Reviewer #3:

Remarks to the Author:

General Comments:

Overall, I think this is an extremely important topic for the remote sensing and ecological/carbon mapping community to be made aware of. There is a large amount of ignored error propagation being published and it needs to be addressed. These authors make a compelling case that significant relationships can be explained largely by spatial proximity rather than any underlying physical attributes (e.g. nematode abundance). However, I find serious limitations to the methods and datasets the authors have presented and have some suggestions for a more informative way of presenting their study, assuming I have gleaned their research objectives correctly.

First, I am genuinely left wondering why the authors are utilising coarse resolution, optical imagery for mapping above ground biomass, even if it is just meant to illustrate its poor predictive power. They do provide a few references dotted around in their discussion and results that have argued that optical data is very poor at capturing structural attributes related to biomass; however, it seems that this point is pretty critical to this piece but is not prominent enough. Their introduction hardly touches on what backscatter or reflectance of satellite imagery is even attempting to capture when used to produce wall to wall maps of forest biomass. Instead, and I understand there are word constraints in these articles, the prose primarily focuses on the statistical limitations of previous efforts because they have not taken into account spatial autocorrelation. I appreciate this last point is the primary focus of the article, but one would also assume that improving biomass mapping would also rely on technology that would actually be capable of interacting with the structural attributes that would be directly

related to carbon storage (e.g. lidar or synthetic aperture radar). Therefore, focusing this piece on a satellite technology known to have poor predictive power seems like an odd choice. If the authors have chosen MODIS because they feel it is also being relied on too heavily for these types of mapping efforts, then this needs to be made more explicit throughout their manuscript. Further, upon reading the methods, it is revealed that the MODIS layers used are composites of 10-year averages. Why was this decision made? I am not sure how meaningful a reflectance value for a specific wavelength (e.g. SWIR) averaged from 8-day composites over a decade would really be. Again, if this is common practice in the literature, please make this explicit, because it does not make a lot of physical sense to me.

Secondly, the underlying field data that the authors are utilising for validation is a newly published dataset in its own right. I am grateful the authors have included the description of this dataset with their submission, because they did not provide an actual link to the data on their form. Perhaps they forgot to, but there was only a blank space after the words "The 1-km resolution raster of forest biomass estimations is available for review purpose on the following temporary link : ." While this dataset is indeed impressively large, it has its own limitations, as outlined by the attached document to this submission. The data presented in Figure 1 has been extrapolated from several commercial forest inventory plots using statistical relationships derived from diameter size classes observed in spatially disparate scientific plots. The attached document suggests that as the data is aggregated at large spatial scales, the overall error reduces; however, as this piece is dealing with the importance of not extrapolating locally relevant forest characteristics to different forest areas, I wonder how the authors would justify the methods used in generating this dataset. This is not meant to suggest that the underlying data is too unreliable to be used in the way they have, but perhaps some discussion of how differences in disturbance history or edaphic features may impact the distribution of individual trees across size classes as well as delving into some of the ecological aspects that may make comparing protected forest areas, presumably where scientific plots are located, and timber concessions problematic. Perhaps the authors could present any differences in plot-averaged wood density or height that may be varying across regions, which has been found to be an important component to the Amazon's above ground biomass gradient? Also, in their methods they state that they only extracted pixels from Figure 1 data that had at least 3 *0.5 ha plots per 1 km². It would help the reader if Figure 1 included a mark (like an "x") indicating which pixels were not included or conversely identifying which ones were. As it is, this Figure gives the impression that there already is reliable wall to wall biomass mapping in various forest fragments in this region and I am unclear if that is the case.

Thirdly, regarding the methods used, I believe I understand what the authors did for their first two random forest-based strategies; however, I am less clear how they ran their LOO method. The authors state that they only wanted to consider the predictive power of a model for spatially relevant data, therefore, am I right in assuming that they only tested predictions for pixels within the same of 44 identified clusters but beyond 0 and 150 km, depending on the iteration being run? Or when they state that "optical and environmental conditions at the test location still fell within the range of values found in model's calibration domain", could the model be assessing pixels at a different location spatially that exhibited similar climatic or reflectance measures? Also, they state they went through 10 km iterations between 0 and 150 km for 10 iterations, but wouldn't that be 15 iterations? The paragraph describing these methods is a bit hard to follow. Which also brings me to the authors' use of WorldClim data. While a commonly used dataset in the literature, it is not very reliable for Africa and certainly not for the interior of Africa. It would be hard to imagine that this dataset would offer much in term of differentiation across the Congo Basin, for instance. A process-based reanalysis product, like ERA-5 might give more reliable metrics, rather than one based on statistical interpolation across a sparse station network. However, having said that, it is not clear from the methods over what time period these climate variables have been averaged. As these are predominantly mature forest areas, an average climate condition for the last few decades may not be particularly relevant to the current forest structure; hence, being able to take a longer perspective, which WorldClim may allow,

might be preferable? In any case, this could be discussed more directly.

The discussion around the limitations of previous maps based on GLAS analysis, due to the close proximity of sampled points along orbital lines, is interesting, but it does not address the superiority of Lidar data in assessing forest structure relative to optical data like MODIS. It is not clear why the debate needs to be an either/or (e.g. lidar vs optical data) rather than a more constructive discussion of how to enhance our ability to produce reliable above ground biomass maps of vegetation. The conclusion of the article is correct and important that going forward machine learning models need to take into account the spatial distribution of the underlying data, but I also wonder if the conclusion could relate to the spatial distribution of data we already have available. Would the authors suggest that above ground biomass measures are still too sparse? Is there a potential for greater use of commercial timber company data to improve existing models, for instance, or should the scientific community be collaborating to establish more spatially independent forest plots for long-term monitoring? I believe the authors could be making more ambitious recommendations.

As I said above, I do believe this piece illustrates important caveats for the scientific community to consider, but I think it really needs a major revision to be of the calibre required for this journal. Personally, I am not convinced of the choice of underlying datasets used and, therefore, would suggest that the authors at the least provide a stronger justification for why they have been chosen if not utilise more physically relevant spatial datasets. I provide more specific comments in the next section.

Specific Comments:

Line 41: "spoil" is an odd word to use here. I would suggest "challenged" or "limited".

Figure 1: A gradient of red to green is not considerate of the majority of colour-blind individuals. I would recommend a different colour scheme.

Line 156: I am not sure what "alimentering the debate over the utility of MODIS data in this context" means.

Lines 186-189: This is a really nice contrast to present. It is very convincing.

Lines 194-196: Again, really nice evidence presented.

Lines 283-284: This sentence seems to be subtly referencing the documented limitation of optical data for mapping biomass in dense forest and, I would argue, needs to be more prominently presented in this manuscript.

Lines 295-302: This is very powerful evidence that may benefit from a bit more discussion earlier on.

Line 325: Not very encouraging to use the terms "seemingly relevant global map layers", this makes it sound like your analysis was more opportunistic than hypothesis driven. This seems consistent with the a lot of text in this submission, which focuses more on the statistical strength of analyses rather than any expected relationships to be observed.

Lines 329-332: As mentioned, the authors do not state over what time period these monthly averages have been calculated.

Lines 345-346: Unsure what "and opposed Arenic Acrisols to a mixed class of other soil types" means.

Line 352: I am surprised that EVI and NDII indices were calculated after taking reflectance averages. I

would have calculated the indices for each relevant composite and then filtered out the pixels using quality control bands. I understand that this method was already published, but it does not seem physically defensible to me.

Table 1: It would be helpful to include the mean and range of above ground biomass values used. Also, it is evident from this table that the climate variables showed very small ranges. Whether that is related to the poor reliability of WorldClim data or just the small amount of variation that exists across the Congo Basin is difficult to say.

Lines 367-369: I am not clear if the removing of pixels that were not consistently forest over the study period was because the ground data would suggest they should have been (as in not impacted by logging or other activity) and this would be a commission error on the part of Hansen et al?

Line 371: I am not sure the reference chosen supports the statement being made. This reference seems to be pushing back on the specific analysis Baccini et al were presenting rather than stating some fundamental or consistent mismatch between Landsat and MODIS.

Line 387: Refers to the use of a fire dataset, but there is no other mention of where this might have been used.

Line 413: "Withholded" should be replaced by "withheld".

Lines 412- 432: As I mentioned in my general comments, I had difficulty following the description of this method. I hope my clarifying questions above make clear where I was confused.

Figure 5: It may help in clearing up my confusion, and perhaps similarly confused readers, if your illustration figures for each strategy showed if each collection of spots were in the same k-cluster or not. Because judging by the contrast in dot colours between the 2nd and 3rd strategies, I would assume that training models were not grouped by k-clusters.

RESPONSE TO REVIEWERS

REVIEWER #1.

General comments.

1. *This is an excellent, important, and much-needed study that reveals central problems with commonly used algorithms for mapping forest biomass and other ecosystem variables. Essentially, it shows that the apparent high predictive power of these algorithms is due to the way cross-validation is conducted, with validation points close in space to training points, to spatial autocorrelation in the response variable, and the fact that combinations of the predictor variables can be used as proxies for location. Many high-profile results need to be reconsidered in light of these findings. My only concerns are fairly minor, and are in regard to some areas where I believe the methods and/or results need to be better explained to the reader.*

We are pleased that you share our opinion on the importance of the problem raised in this study.

2. *As the authors surely know, spatial structure in independent variables and response variables alone (of the kind shown in figures 2a,b) is not necessarily a problem for most statistical analyses – what is a problem is spatial autocorrelation in residuals. The graphs in figure 2a,b and associated text are potentially misleading in this sense, as they focus on spatial autocorrelation in the variables. Figure 3c shows the spatial autocorrelation in the residuals, which is the real problem, and also that these appear uncorrelated in the typical (problematic) cross-validation approaches. The text explaining this figure and its significance could be clearer; these are important points to get across and deserve some more explanation (lines ~190-200)*

Figure 2 (now split into Figures 2 and 3) aimed at providing readers some insight on how our set of reference biomass data is distributed in space (Figure 3 panels c, d, e) and on the ranges of autocorrelation in the variables (Figure 2 panels a, b). In this associated text, we lay out what the issue is from a cross-validation perspective: given (i) the range of autocorrelation in the variables and (ii) the spatial density (or proximity) of reference biomass data, randomly selecting a pixel (or a group of pixels) for model validation will not constitute an “independent” test set because numerous pixels in its neighborhood have very similar characteristics (both in terms of the dependent and independent variables). Throughout the manuscript, our main focus is put on this issue of model (cross-)validation.

Spatial autocorrelation in model residuals is a separate issue. Reducing spatial autocorrelation in the residuals aims at removing any bias in the estimates and optimism in the parameters’ standard errors. However, even a model without any residual spatial autocorrelation cannot be validated correctly for its predictive ability when the test data are not spatially independent. It is the spatial proximity of train and test data that causes the problem in model validation, and it requires a spatially non-random (blocked) validation data set.

The two issues (i.e., autocorrelated residuals, and model validation in a spatially structured environment) are distinct, but are not completely independent, in that using a ‘*typical (problematic) cross-validation*’ on spatially structured data may provide biased (over optimistic) estimations of model residuals (to the point that they may not show any spatial structure, as in Figure 2-c).

We strengthened the description – and distinction – of those two separate issues in the text associated to Fig. 2 (now Figures 2 and 3, L. 117-125, and 136-139), which should also make clearer

that what this study addresses is the effect of spatial autocorrelation on model validation (regardless of the presence of spatial structures in model residuals). In Figure 3-c (now Figure 2-c), which does depict spatial autocorrelation in models residuals, we merely illustrate the fact that an inappropriate model validation design (associated to what we refer as ‘the second issue’ in L. 122) may in fact hide the presence of autocorrelation in model residuals (i.e., what we refer as ‘the first issue’ L. 117), which further stress the importance of considering data spatial autocorrelation in model validation.

3. *I struggled to figure out exactly what was being done in some of the analyses, and was surprised not to be able to find some details even in the SI. In particular, I could find nothing in the main text to explain how figure 4c is constructed, and it was hard for me even to figure out the explanation in the SI.*

There was indeed no information on how Figure 4-c (now Figure 5-c) was constructed. In the new Fig. 5-c, we simply projected the fitted relationship observed in Fig. 5-a on the study area. For each forest pixel in the map, we calculated the distance to the closest reference AGB pixel, and retrieved the corresponding mean R^2 from the fitted relationship illustrated in Fig. 5-a. We modified the figure’s caption to provide this information (L. 257-259).

4. *It also took me a while to understand the results shown in figures 4a,b. (At first I thought that the cross-validation was as usual being done on a random 10% of the data, and that the analyses were leaving out ever more data around that 10% from the testing data, so I wondered if the result was an artefact of having less and less testing data to work from. Then finally I reached the text late in SI explaining that only one point was evaluated at a time, but only after I’d been wondering about this for a while...)*

We apologize for the misunderstanding and made this clearer in the revised manuscript. The two types of cross-validation used in this study (i.e. K-fold and Leave-one-out cross-validation) are introduced and briefly described at the end of the Introduction section (L.166-178). In particular, the buffered leave-one-out cross validation (B-LOO CV) used to generate Figure 4 a-b (now Figure 5 a-b) is described as being “similar to a leave-one-out cross-validation (i.e. where the test set consists of a single observation)”. To avoid any misunderstanding, we modified the Results paragraph associated to this figure (L. 225-233) to make clear how the B-LOO CV works. We believe that this modification will solve the problem you faced and – by contrasting the B-LOO and spatial K-fold CVs (see also our response to comment n°5) – it better explains why we did perform a B-LOO CV in the first place.

5. *In terms of interpreting results, how do you explain that the r^2 values in figure4a drop even below those from the spatial cross-validation? Does this mean that the performance of the spatial cross-validation is itself still inflated by spatial autocorrelation across the borders of the patches that are included or dropped?*

Yes, you are fully right. The spatial K-fold CV allows increasing the average spatial distance between observations in the training set and observations in the test set. This “average spatial distance” is directly related to the size of the spatial clusters: if clusters are too small, some residual correlation between training and test sets remains, and statistics of spatial CV would still be inflated (see for instance Box 1 of Roberts *et al.*, 2017, *Ecography*, and associated appendix). Using the B-LOO CV allows going one step further than the spatial K-fold CV (which, however, has a large computational cost) in that it allows assessing model predictive error for a range of distances between training and test data sets. In Figure 4-a (now Figure 5-a), our results indeed indicate that CV statistics obtained using spatial K-fold CV are still optimistic (i.e. a larger cluster size should be used to obtain an estimate of model predictive power closer to the true error).

We added a horizontal line to this figure representing the R^2 of the spatial K-fold CV, and amended the text of the corresponding Results' paragraph to underline this aspect (L. 244-246).

6. *The authors note that overfitting is not as big a problem when a generalized linear model is used (265-267). Personally, I would be very interested to see that parallel result. I hope the authors intend to publish that somewhere, even if not here.*

We added a figure in Supplementary material (Fig. S1, reported below as Fig. R1) to illustrate this statement. Using a Partial Least Squares Regression (PLSR) instead of Random Forest (RF) for example, we can observe a different evolution pattern for the R^2 in the B-LOO-CV: the R^2 is low (poor model predictive power) right from the start (i.e. without exclusion buffer), and remains "low" as buffer radius increases. With radius of 150 km, i.e., when spatial autocorrelation is not anymore an issue, the two methods provide similar accuracy. Hence, PLSR does not overfit the data as much as RF in presence of spatial autocorrelation.

Fig. R1. Influence of data spatial structure on model's CV statistics. a, Evolution of the coefficient of determination (R^2) between predicted and observed pixels' AGB as buffer radius for neighboring pixels exclusion increases in the buffered leave-one-out cross-validation (B-LOO CV). Predictions are made with the Random Forest model based on MODIS and environmental variables (RF_{FULL}, red) and a Partial Least Squares Regression (PLSR) model based on the same set of predictors (PLSR_{FULL}, green). b, Evolution of the root mean squared prediction error (RMSPE) in the B-LOO CV. In addition to RF_{FULL} and PLSR_{FULL}, the RMSPE of a null model that systematically predicts the mean of training data is plotted (Model_{NULL}, black).

Additional specific comments.

7. *Figure 2. It might be better to split this into 2 or more separate figures. In particular, it would be nice to see one figure devoted exclusively to the semivariograms, and to have that figure better show the pattern at relatively short spatial scale. For example, in figure 2a, the scale is such that it is hard to tell at what distance the semivariance of AGB (dark red line) asymptotes. Many empirical ecologists will be interested to know the shape of the relationships in the first 250 km in greater detail.*

Thanks for this suggestion. We did split Figure 2 in two, grouping all variograms in a single Figure, as suggested. We also improved variograms readability by increasing their size, having a finer trend line, and adding a grid in the figures' background.

It should be noted, however, that spatial autocorrelation is dependent on the resolution of the data. For forest biomass, the range of spatial autocorrelation is usually smaller than 10 km when assessed at the c. 0.5-ha plot scale (e.g. Guitet *et al.*, 2015, *PLOS one*), while aggregating data at larger scale smooths spatial patterns and increases spatial autocorrelation (e.g. this study). Our results are thus relevant for a particular spatial resolution, 1 km, which may not be the most common resolution most ecologist work at.

8. *Figure 2c requires a distance scale.*

A scale is now added to the Figure.

9. *Lines 238-239. It would be interesting to know a bit more about the fitted RF models. Is there any way the general issues could be diagnosed from the structure of the RF models? For example, if the models in part take the form of a decision tree, could one map out the branches of the decision tree and see that the combinations are essentially proxies for spatial location?*

Unfortunately, we do not see any way to produce a convincing (and concise) diagnosis of the RF internal structure that would illustrate the issue. One difficulty is that it is possible to graphically illustrate a single tree, but as far as we are aware, we cannot map the entire forest on which the model bases its predictions. Besides, the issue at hand stems from spatial autocorrelation in the data. Spatial autocorrelation is about distance (between sample points), not location. So, one would need to visualize whether predictors vary in a specific way as the distance between points changes. In practice, a strategy could be to (i) train the model, (ii) predict forest biomass keeping all predictors but one at their median value, and then (iii) use a semivariogram on model's predictions to check for spatial structuring: if there is a pattern, the predictor that was left to vary may indeed act as spatial distance proxy. This would not be a direct diagnosis of model internal structure as suggested, and we are not convinced that it would provide more conclusive evidence of the problem at hand than what's already presented in the manuscript.

10. *Lines 254-55. Saying all points within 100 km are pseudoreplications seems overstated... the key point is not the spatial autocorrelation of the independent or dependent variables, but of the residuals. That seems to be somewhat shorter, isn't it? It would be nice to see figures 2a,b, and figure 3c in the same figure, and to see the first part (<200 km) of each magnified to better see at what distance the semivariance asymptotes in each case.*

Regrettably, our statement is correct. As mentioned in our answer to comment n°2, the spatial autocorrelation of the *raw* data is crucial for validation: the test data set must be independent of the training data, irrespective of the model used. Hence, it cannot be the model residuals that determine the range of problematic autocorrelation. Please see also the response to comment n°7 for the issue on figures.

11. *Table 1. What makes the mean temperature annual? That is, isn't it just mean temperature over some time period? Same for other "annual" variables. For seasonality, need to specify somewhere the time scale over which standard deviations and CVs of temperature and precipitation are calculated. Often, for example, it is the standard deviation among months of monthly mean (over years) temperatures. MODIS reflectance – specify units. Topography – specify the spatial grain of the slope, convexity, etc. That is, the same point can have a different slope if it is the slope at 1*

km scale vs. 100 m vs. 10 m. And similarly, a point can be on a small ridge (10 m scale) within a valley (1 km scale), so need a spatial grain for convexity.

Concerning climatic variables, the WorldClim database provides (i) monthly means of climate variables and (ii) a set of 19 standard ‘bioclimatic’ variables derived from (i) (see here). Bioclimatic variables include, for example, the ‘annual mean temperature’ computed as the mean of the 12 monthly temperature means. In the manuscript, we used both the terminology and computing approach of WorldClim. We clarified the text to make this clearer (L. 384-389).

Concerning topographical variables (*‘slope, convexity, etc’*), we used a 30-m spatial grain (i.e., the resolution of SRTM data) (L. 399-401). Note that we corrected a mistake in the initial version of this sentence, where we wrongly specified a 90-m resolution.

MODIS reflectance is unitless. We homogenized our descriptions of all unitless variables by adding the term ‘unitless’, between brackets, in the *Description* column of Table 1 (as previously done for the Water Availability (WA) variable, for instance).

12. 421. *Would it make sense to repeat this for different cluster sizes?*

Repeating the spatial K-fold CV with different cluster sizes would lead to similar results to those of the B-LOO CV: it would show a decrease in model CV statistics as cluster size (or ‘the average distance between training and test data sets’) increases. There would be fundamentally nothing wrong in doing that, but the B-LOO CV offers a better control on the distance between training and test data sets (as opposed to ‘an average distance’ in the spatial K-fold CV strategy).

13. 440-445. *Great safeguard!*

Thanks.

14. 449-451. *I recommend making the code used for this analysis publicly available at time of publication.*

We added a ‘Code availability’ section, specifying that the code is available upon request from the corresponding author. While we do not mind sharing this code, an R package has recently been released (see reference below) and offers a set of functions to perform random and spatial cross-validations, buffered leave-one-out cross validation along with other useful tools to design, visualize and perform model validation in a spatially structured environment. This package not only covers the analyses we did in this study, but may also provide more computationally-efficient functions. We believe our code, specific to a given study case, would not add much to what is already available to users out there. We added the R package’s reference to the manuscript (L. 166).

Valavi, R., Elith, J., Lahoz-Monfort, J. J., & Guillera-Arroita, G. (2018). blockCV: an R package for generating spatially or environmentally separated folds for k-fold cross-validation of species distribution models. bioRxiv, 357798.

REVIEWER #2.

General comments.

15. *Overall, I think this is an extremely important topic for the remote sensing and ecological/carbon mapping community to be made aware of. There is a large amount of ignored error propagation being published and it needs to be addressed. These authors make a compelling case that significant relationships can be explained largely by spatial proximity rather than any underlying physical attributes (e.g. nematode abundance). However, I find serious limitations to the methods and datasets the authors have presented and have some suggestions for a more informative way of presenting their study, assuming I have gleaned their research objectives correctly.*

Thank you for your constructive remarks. Your review suggests that there are some important points in the Introduction section that do not sufficiently stand out, making the context – and the objective – of this study unclear. We tried to address these shortcomings in our responses to your comments as well as in the revised version of the manuscript.

16. *First, I am genuinely left wondering why the authors are utilising coarse resolution, optical imagery for mapping above ground biomass, even if it is just meant to illustrate its poor predictive power. They do provide a few references dotted around in their discussion and results that have argued that optical data is very poor at capturing structural attributes related to biomass; however, it seems that this point is pretty critical to this piece but is not prominent enough. Their introduction hardly touches on what backscatter or reflectance of satellite imagery is even attempting to capture when used to produce wall to wall maps of forest biomass. Instead, and I understand there are word constraints in these articles, the prose primarily focuses on the statistical limitations of previous efforts because they have not taken into account spatial autocorrelation. I appreciate this last point is the primary focus of the article, but one would also assume that improving biomass mapping would also rely on technology that would actually be capable of interacting with the structural attributes that would be directly related to carbon storage (e.g. lidar or synthetic aperture radar). Therefore, focusing this piece on a satellite technology known to have poor predictive power seems like an odd choice. If the authors have chosen MODIS because they feel it is also being relied on too heavily for these types of mapping efforts, then this needs to be made more explicit throughout their manuscript.*

The context and objective of our study is given in the first paragraph of the manuscript. In light of your comment, we rephrased this paragraph, making our study's objective more explicit (L. 59-61). In a nutshell: the two pantropical forest biomass maps of Saatchi et al. (2011, *PNAS*) and Baccini et al. (2012, *Nat. Clim. Chang.*) are being widely-used by the scientific and decision-making/practitioners communities; both studies have reported high predictive power for their mapping models, yet the two maps show strong disagreements (between each other and with higher quality local maps). This contradiction has been the source of multiple studies, aiming at understanding the source of maps disagreements. In the present study, we draw attention to a methodological flaw in the validation schemes of those pantropical maps that we believe largely explains the said contradiction. Our study is grounded in the thematic of 'large-scale biomass mapping', but the methodological flaw we evidence applies well-beyond this thematic.

In order to evidence the issue, we reproduced the general modelling approach used in pantropical biomass mapping studies (L. 62-65), which naturally entails using the same type of remote-sensing data (i.e. MODIS) and the same processing steps (cf. your comment n°17) – regardless of their

relevance for the purpose of biomass mapping. Again, our objective here is not to ‘*improve biomass mapping*’ (for instance using ‘*technology that would actually be capable of interacting with the structural attributes that would be directly related to carbon storage*’) – but rather to illustrate, with a concrete example, an important methodological flaw: even with ‘*satellite technology known to have poor predictive power*’, the standard approach for model validation suggests that the model is predictive.

That being said, from a biomass mapping perspective, we completely agree with the comments you make across your review, notably on the need to use spaceborne signals with clearer biophysical relationships with biomass, or parsimonious models with sound (hypothesis-based) variable selection. We believe that in order to move toward such studies aiming at ‘*improving biomass mapping*’, we first need to improve and formalize the way we evaluate biomass mapping models (which is a subject on its own, that we address in this study).

17. *Further, upon reading the methods, it is revealed that the MODIS layers used are composites of 10-year averages. Why was this decision made? I am not sure how meaningful a reflectance value for a specific wavelength (e.g. SWIR) averaged from 8-day composites over a decade would really be. Again, if this is common practice in the literature, please make this explicit, because it does not make a lot of physical sense to me.*

Using MODIS composites averaged over years indeed is the approach followed in Baccini et al., 2012 (Nat. Clim. Chang.). In a following publication (Baccini et al., 2017, *Science*), the authors projected the model on yearly MODIS composites and claimed that it allowed reliably monitoring, across the tropics, subtle annual changes in biomass related to forest degradation and growth. We thus followed the same approach (cf. our response to comment n°16). In practice, we generated a 10-year composite (as stated in the Methods, but also in the Introduction section, i.e. L. 97) over 2000-2010 so as to match the period of field data collection. The text of the Methods was modified to make this clearer (L. 409).

18. *Secondly, the underlying field data that the authors are utilising for validation is a newly published dataset in its own right. I am grateful the authors have included the description of this dataset with their submission, because they did not provide an actual link to the data on their form. Perhaps they forgot to, but there was only a blank space after the words “The 1-km resolution raster of forest biomass estimations is available for review purpose on the following temporary link : .”*

This was indeed a mistake, our apologies. The dataset can be accessed here:

<https://figshare.com/s/26fad5660e34d89966a7>

19. *While this dataset is indeed impressively large, it has its own limitations, as outlined by the attached document to this submission. The data presented in Figure 1 has been extrapolated from several commercial forest inventory plots using statistical relationships derived from diameter size classes observed in spatially disparate scientific plots. The attached document suggests that as the data is aggregated at large spatial scales, the overall error reduces; however, as this piece is dealing with the importance of not extrapolating locally relevant forest characteristics to different forest areas, I wonder how the authors would justify the methods used in generating this dataset. This is not meant to suggest that the underlying data is too unreliable to be used in the way they have, but perhaps some discussion of how differences in disturbance history or edaphic features may impact the distribution of individual trees across size classes as well as delving into some of the ecological aspects that may make comparing protected forest*

areas, presumably where scientific plots are located, and timber concessions problematic. Perhaps the authors could present any differences in plot-averaged wood density or height that may be varying across regions, which has been found to be an important component to the Amazon's above ground biomass gradient?

There seems to be a misunderstanding on how the data presented in Fig. 1 have been computed. Biomass estimations on CoFor data (i.e. the set of c. 190,000 management forest inventory plots used to generate Fig. 1) are unrelated to any 'statistical relationships derived from diameter size classes observed in spatially disparate scientific plots'. The CoFor dataset indeed provides all necessary information to estimate tree and plot biomass, including tree diameter measurements, tree taxonomic identification (used to derive wood density), plot size, etc. In the accompanying manuscript, scientific plots are used to get some sense of the estimation error on 0.5-ha plot biomass estimations that can be expected from the computation scheme applied to CoFor data. For example, tree diameter measurements in CoFor data are provided as discrete diameter classes rather than continuous values. We thus developed a 'specific function' in order to assign a continuous diameter to each tree from its diameter class. In order to assess the error on biomass estimations associated to this computation step, we discretized the continuous tree diameters available in scientific plot data, applied the diameter-assignment function, and compared the resulting biomass estimations to the ones obtained from raw, continuous diameters. Scientific plots are thus used to evaluate the computation scheme, but do not impact CoFor biomass estimations. Any systematic difference between the CoFor and scientific datasets (be it in terms of disturbance history, edaphic features, average wood density, etc.) would at most bias our assessment of biomass estimations error, but would not impact biomass estimations themselves.

It should be noted that this accompanying manuscript is itself accepted for publication in *Scientific Data*. The final version of the accompanying manuscript include a comparison of the range of forest structure parameters covered by both the CoFor and scientific datasets (see below). This Figure shows that forest structure in the scientific dataset covered most of the structural range found in CoFor, at the exception of forests characterized by both a small mean tree size and a low tree abundance, likely reflecting degraded forest states that can notably be found in Northern Congo (the so-called *Marantaceae* forests).

Figure R2. Range of forest structural parameters in 0.5-ha plots from the CoFor (heat color gradient) and scientific (black crosses) datasets. $N_{\geq 20}$ and $D_{g \geq 20 \text{cm}}$ stands for the number of trees and the

quadratic mean tree diameter for trees with diameter at breast height greater or equal than 20 cm, respectively.

In the present manuscript, we edited the Methods section to clarify the fact that scientific plot data were only used to evaluate the biomass computation scheme (L. 369-370).

20. *Also, in their methods they state that they only extracted pixels from Figure 1 data that had at least 3 *0.5 ha plots per 1 km². It would help the reader if Figure 1 included a mark (like an “x”) indicating which pixels were not included or conversely identifying which ones were. As it is, this Figure gives the impression that there already is reliable wall to wall biomass mapping in various forest fragments in this region and I am unclear if that is the case.*

Including “a mark (like an “x”) indicating which pixels were not included or conversely identifying which ones were” in Fig. 1 is not possible, because the spatial resolution of biomass pixels in Fig. 1 is 5 km – not 1 km. We indeed omitted to report this detail in the Figure’s caption, which has now been corrected. Aggregating data to a 5 km pixel grid in this Figure serves a purely visual purpose: given the regional extent the data covers, representing biomass pixels at the 1 km resolution makes it hard for readers to actually see the biomass level within pixels. An example of this Figure with biomass pixels resolution set to the 1 km is given below to illustrate the issue (Fig. R3-a). For the reviewer’s point, we however agree that a map representing which 1 km pixels have been selected / discarded for the analysis would be useful to the reader, and we added such map (Figure R3-b) as a Supplementary Figure (S4).

Fig. R3. (a) Overview of study area and field data distribution (i.e. Fig. 1 of the manuscript) with biomass pixel resolution set to 1 km. (b) Spatial distribution of CoFor-AGB 1 km biomass pixels, with pixels filtered out from the dataset in black and pixels used for biomass modelling in red.

21. Thirdly, regarding the methods used, I believe I understand what the authors did for their first two random forest-based strategies; however, I am less clear how they ran their LOO method. The authors state that they only wanted to consider the predictive power of a model for spatially relevant data, therefore, am I right in assuming that they only tested predictions for pixels within the same of 44 identified clusters but beyond 0 and 150 km, depending on the iteration being run? Or when they state that “optical and environmental conditions at the test location still fell within the range of values found in model’s calibration domain”, could the model be assessing pixels at a different location spatially that exhibited similar climatic or reflectance measures?

This comment, together with comments n°22 and 44 and 45, indicates that our description of the B-LOO CV, and perhaps the associated figure (now listed as Figure 6) were not sufficiently clear. We thus deeply modified the text and the figure.

In the B-LOO CV, just like in the typical leave-one-out CV, a single test observation is left-out for testing per model run. Unlike the leave-one-out CV though, a buffer is used to remove training observations at the neighborhood of the test observation, and that for different buffer sizes (or buffer radii r). Here, we used 16 r values: from 0 to 150 km by 10 km, meaning that the model was trained and validated 16 times per test observation. In order to compute one estimate of model's R^2 and RMSE, we performed the B-LOO CV on 100 randomly selected test pixels across the entire dataset (without any consideration to any 'cluster'), meaning that the model was trained and validated 1,600 times. Last, in order to generate the average R^2 and RMSPE displayed in Fig. 5 a-b, we iterated this procedure 10 times (i.e. 16,000 model runs in total).

Removing a single test observation from a dataset of size $> 20,000$ does not impact much model's training set. However, for large r values (e.g. $r = 150$ km), a substantial number of observations at the neighborhood of the test observation are removed from the training set (up to c. 10% of the full data set). Since all removed data are closely grouped in space (within the buffer), a risk is to remove an entire portion of the environmental (or reflectance) domain from the model training set. In these conditions, using the model to predict biomass at the test location, outside the model calibration domain, would amount to a case of model extrapolation (leading to an inflation of prediction error independent of the problem at hand). In order to avoid model extrapolation, we made sure each randomly selected test pixel was still within the model's calibration domain even when removing all its neighbors at the largest r values (i.e., we verified that test pixel covariate values fell within the range of model calibration domain).

22. *Also, they state they went through 10 km iterations between 0 and 150 km for 10 iterations, but wouldn't that be 15 iterations? The paragraph describing these methods is a bit hard to follow.*

See our response to comment n°21.

23. *Which also brings me to the authors' use of WorldClim data. While a commonly used dataset in the literature, it is not very reliable for Africa and certainly not for the interior of Africa. It would be hard to imagine that this dataset would offer much in term of differentiation across the Congo Basin, for instance. A process-based reanalysis product, like ERA-5 might give more reliable metrics, rather than one based on statistical interpolation across a sparse station network. However, having said that, it is not clear from the methods over what time period these climate variables have been averaged. As these are predominantly mature forest areas, an average climate condition for the last few decades may not be particularly relevant to the current forest structure; hence, being able to take a longer perspective, which WorldClim may allow, might be preferable? In any case, this could be discussed more directly.*

We edited the Methods (L. 382) to specify the timeframe over which WorldClim climate variables are computed (comment n° 35).

As far as we know, there are large uncertainties on all wall-to-wall climate products over central Africa due to a lack of climate stations. This lack of stations indeed makes WorldClim data uncertain. For the same reason, the potential superiority of alternative products cannot be properly assessed. Here, we used WorldClim because of its widespread use in the literature (notably to study tropical dense forest AGB variation, e.g., M.J.P. Sullivan *et al.*, 2020, *Science*) and its 1 km spatial resolution (which coincides with the resolution of other datasets used in this study). Your concern over the

relevance of this product echoes comment n°16 (i.e. the relevance of MODIS). Although we share the underlying idea, namely that alternative variables (or modelling approaches for that matter) could have led to some improvement of model predictive power, the objective of this study is not to ‘*improve biomass mapping*’ (see our response to comment n°16). Rather, our objective is to illustrate a flaw in the most common approach of mapping model’s validation. The issue could have been shown with simulated data – instead, we used commonly employed data, which we believe make the demonstration much more striking to the readers. Again, we firmly believe that building awareness on this fairly methodological issue has a much greater potential impact on future studies (and *in fine* on our understanding of forest biomass distribution) than doing another study case on biomass mapping.

24. *The discussion around the limitations of previous maps based on GLAS analysis, due to the close proximity of sampled points along orbital lines, is interesting, but it does not address the superiority of Lidar data in assessing forest structure relative to optical data like MODIS. It is not clear why the debate needs to be an either/or (e.g. lidar vs optical data) rather than a more constructive discussion of how to enhance our ability to produce reliable above ground biomass maps of vegetation.*

Several issues are raised in this comment.

First, although there is no doubt that spaceborne LiDAR data (GLAS, GEDI or MOLI) is more sensitive to forest biomass gradients than optical data, these LiDAR datasets provide a discrete sampling. As a consequence, biomass estimations at LiDAR sampling locations should then be extrapolated with wall-to-wall remote-sensing data (e.g., MODIS) to generate biomass maps, as done with GLAS data in previous large-scale biomass mapping studies (Saatchi et al. 2011, Baccini et al. 2012, 2017). Hence, from a methodological point of view, the reference set of LiDAR-derived biomass estimations in previous studies can be compared to our reference set of field-derived biomass estimation (i.e. CoFor data), both of them being then extrapolated using optical data. The superiority of LiDAR data over optical data for assessing forest biomass, that we fully acknowledge, is thus irrelevant in the context of the methodological framework developed here to test validation methods of wall-to-wall extrapolation models (i.e. whatever the initial source of data). We modified the text to make this methodological issue stand out more clearly (L. 74-76).

Second, we believe that “*discussion around the limitations of previous maps based on GLAS analysis, due to the close proximity of sampled points along orbital lines*” is crucial. Indeed, our study ‘mimics’ but is not an exact ‘replicate’ of pantropical biomass mapping studies. For instance, we used field-derived and not GLAS-derived reference biomass estimations, among other differences. In this part of the Discussion, we are making the case that our results (i.e. an over-estimation of model predictive power when using 10-fold random cross-validation) can be generalized to previous studies, despite the aforementioned differences.

Last, although we understand the usefulness of a ‘*constructive discussion of how to enhance our ability to produce reliable above ground biomass*’ – for instance using remote-sensing data types having greater sensitivity to biomass variations than optical data, or using mapping models with clearer biophysical bases, we consider that it is not fully within the scope of this paper (cf. our clarification on this study’s objective, comment n°16), which must be kept short and focused to conform to the journal format.

25. *The conclusion of the article is correct and important that going forward machine learning models need to take into account the spatial distribution of the underlying data, but I also wonder if the conclusion could relate to the spatial distribution of data we already have available. Would*

the authors suggest that above ground biomass measures are still too sparse? Is there a potential for greater use of commercial timber company data to improve existing models, for instance, or should the scientific community be collaborating to establish more spatially independent forest plots for long-term monitoring? I believe the authors could be making more ambitious recommendations.

From the standpoint of large-scale biomass mapping, the most promising perspective comes from the recent launch of the GEDI mission as well as the upcoming BIOMASS mission. GEDI (a spaceborne LiDAR sensor) for instance will provide billions of discrete biomass estimations across the tropics for the period 2020-2022. GEDI-derived biomass estimations, much like the ones from the older GLAS sensor, are better suited to train and validate wall-to-wall AGB models than estimations from scientific field sample plots that are – and most certainly always will be – way too scarce. The reviewer is correct however regarding the fact that existing ground data are insufficiently valorized. Commercial forest inventory, for instance, can be a precious source of information, as shown here (and in Ploton *et al. Scientific Data, in press*). Important financial efforts are also being invested into National Forest Inventories, notably in the frame of REDD+ strategies. A particular attention should be paid to the geopositioning of NFI plots in view of the calibration of remote sensing products (Rejou-Méchain *et al.*, 2019, *Surv. Geophys.*). Last but not least, as significant investments are directed towards developing and launching new sensors dedicated at least in part to AGB estimation, it is crucial in parallel to invest into the deployment and maintenance of high standard scientific plot networks, completed by airborne LiDAR acquisitions (Chave *et al.* 2019, *Surv. Geophys.*). We modified the text (L. 332-343) to emphasize these perspectives and provide a recommendation to the community of remote-sensing scientists.

26. *As I said above, I do believe this piece illustrates important caveats for the scientific community to consider, but I think it really needs a major revision to be of the calibre required for this journal. Personally, I am not convinced of the choice of underlying datasets used and, therefore, would suggest that the authors at the least provide a stronger justification for why they have been chosen if not utilise more physically relevant spatial datasets. I provide more specific comments in the next section.*

See our answers to comments n°16 and n°25.

Additional specific comments.

27. *Line 41: “spoilt” is an odd word to use here. I would suggest “challenged” or “limited”.*

We used “limited” instead.

28. *Figure 1: A gradient of red to green is not considerate of the majority of colour-blind individuals. I would recommend a different colour scheme*

We modified the color scheme to make it easier to read by individuals with colorblindness.

29. *Line 156: I am not sure what “alimentering the debate over the utility of MODIS data in this context” means*

In light of previous comments (notable n°16) we rephrased the introduction to better justify our choice to use MODIS data for AGB mapping. In particular, we stressed that a recent study published in a wide-audience scientific journal (i.e. *Science*) claimed MODIS allowed detecting and monitoring annual changes of forest biomass associated to forest degradation and growth (cf. our response to comment n°17). We also stated that the latter study led to some pushback from other scientists (L. 91-94, reference 23 of the manuscript) questioning “*the relationship (or lack thereof) relating vegetation reflectance to AGB*”. This should clarify what we mean by “*the debate over the utility of MODIS data*” in the context of AGB mapping.

30. *Lines 186-189: This is a really nice contrast to present. It is very convincing.*

Thanks

31. *Lines 194-196: Again, really nice evidence presented.*

Thanks, again.

32. *Lines 283-284: This sentence seems to be subtly referencing the documented limitation of optical data for mapping biomass in dense forest and, I would argue, needs to be more prominently presented in this manuscript.*

As stated in our response to comment n°29, we modified the Introduction to more clearly present the debate on the limitation of MODIS for dense forest biomass mapping. In this study, however, our objective is to clarify the gap between high MODIS-based models’ predictive power on one hand, and poor MODIS-based biomass maps agreement on the other (cf. our response to comment n° 16). Although this objective, and our results, point toward the poor predictive power of MODIS on forest biomass, the latter is not the primary take-home-message of this study. We indeed believe that for such demonstration to be robustly made, one would need to test different approaches to process / use MODIS data, for example exploiting intra-annual variability of vegetation reflectance as a proxy for vegetation phenology, or even directional variations and the link between bidirectional reflectance distribution function (BRDF) and forest structure. Calling for better model validation practices, our hope is that future studies will make this demonstration. Here, we, however, took position in this debate by stating that “*our results likely reflect the well-known saturation of medium and high resolution passive optical data*” (L. 318-319).

33. *Lines 295-302: This is very powerful evidence that may benefit from a bit more discussion earlier on.*

This evidence (i.e. the opening on the Nematodes study), by the end of the Discussion, aims at giving a last support to the main recommendation of our manuscript that we give at the last line of the Discussion section. It also illustrates that the general issue we raised is not restricted to biomass mapping. Since this example refers to a completely different thematic than forest biomass, we find it difficult to mix it with earlier discussion points which addressed previous biomass mapping efforts (and how our results can be generalized to those studies, cf. our answer to comment n° 24).

Line 325: Not very encouraging to use the terms “seemingly relevant global map layers”, this makes it sound like your analysis was more opportunistic than hypothesis driven. This seems consistent with the a lot of text in this submission, which focuses more on the statistical strength of analyses rather than any expected relationships to be observed.

Your interpretation is correct. This formulation is intended as a hint to a more and more widespread modelling strategy, notably in the field of 'Big Ecology' we refer to in the Discussion (e.g. the Nematode study, among others), which consists in using as many 'seemingly relevant' predictors as possible in a data-adaptive machine learning model (with no theoretical assumptions). By stressing how misleading the results of such approaches can be when a (common) flawed cross-validation is used, our study is in fact a critic of this modeling strategy.

34. Lines 329-332: *As mentioned, the authors do not state over what time period these monthly averages have been calculated.*

WorldClim data are based on records from 1970 to 2000 (see here). We added this information to the text (L. 372).

35. Lines 345-346: *Unsure what "and opposed Arenic Acrisols to a mixed class of other soil types" means.*

Soils types from the Harmonized World Soil Database were merged into two categories: Arenic Acrisols (sandy soils) and a mixed category containing all other soil types. We rephrased and developed this sentence to clarify what we meant (L. 402-405).

36. Line 352: *I am surprised that EVI and NDII indices were calculated after taking reflectance averages. I would have calculated the indices for each relevant composite and then filtered out the pixels using quality control bands. I understand that this method was already published, but it does not seem physically defensible to me.*

We indeed followed a published methodological approach (specifically the one of Baccini *et al.*, 2012, *Nat. Clim. Chan.* & Baccini *et al.*, 2017, *Science*). See our answer to comment n°16 for a full response.

37. Table 1: *It would be helpful to include the mean and range of above ground biomass values used. Also, it is evident from this table that the climate variables showed very small ranges. Whether that is related to the poor reliability of WorldClim data or just the small amount of variation that exists across the Congo Basin is difficult to say.*

As requested, we added the mean and range of biomass in the table.

For your information, additional tests with other sources of climate data, notably the Climatic Research Unit (CRU) dataset, lead to fairly similar variables' means and ranges.

38. Lines 367-369: *I am not clear if the removing of pixels that were not consistently forest over the study period was because the ground data would suggest they should have been (as in not impacted by logging or other activity) and this would be a commission error on the part of Hansen et al?*

The idea underlying the exclusion of pixels that underwent some form of disturbance (as detected and mapped in Hansen *et al.*) is to limit undue discrepancies between reflectance values of the 10-y MODIS composite on one hand and the estimation of pixel biomass from field data on the other hand. Taking a caricatural example, if the biomass of a pixel was computed on field-data collected in

2000, and that this pixel was completely deforested in 2001 (showing, hence, the spectral response of bare soil from 2001 to 2010), then the average spectral response of the pixel from 2000 to 2010 would not be representative at all of the 2000's biomass level. This source of variation in the (tested) relationship between MODIS variables and forest biomass estimations emerges from our use of a 10-y image composite and does not reflect the potential sensitivity of MODIS to forest biomass variation *per se*. We thus focus the analysis on pixels with 'stable' canopy cover from 2000 to 2010.

39. *Line 371: I am not sure the reference chosen supports the statement being made. This reference seems to be pushing back on the specific analysis Baccini et al were presenting rather than stating some fundamental or consistent mismatch between Landsat and MODIS.*

The statement being made is that “*there is an influence of canopy cover on the relationship between reflectance variables and forest biomass*”. As you rightfully pointed out in your general comment n°16, the physical basis of the relationship between MODIS reflectance values and forest biomass is unclear/weak. There is, however, a much straightforward relationship between reflectance values and canopy cover (the more vegetated the pixels, the greener). Since fully vegetated pixels tend to have more vegetation biomass than sparsely vegetated pixels, canopy cover somewhat influences the relationship between reflectance variables and forest biomass. The reference that you question backs up this last statement by making the same argument: it states that the “reflectance – biomass” relation on which Baccini *et al.* rely to map forest biomass is essentially the same “reflectance – canopy cover” relation used by Hansen *et al.* to detect deforestation.

In this sentence, we do not imply any “*fundamental or consistent mismatch between Landsat and MODIS*”.

40. *Line 387: Refers to the use of a fire dataset, but there is no other mention of where this might have been used.*

There is a mention of the use of fire dataset at line L. 437-438 of the revised manuscript: “(ii) a non-null percentage of fire occurrence between 2000 and 2012 (736 pixels or 2.5% of the filtered dataset)”. Pixels having experienced fires during the period of interest were filtered out (for the same reason detailed in our response to comment n°38).

41. *Line 413: “Withholdd” should be replaced by “withheld”.*

The correction has been made.

42. *Lines 412- 432: As I mentioned in my general comments, I had difficulty following the description of this method. I hope my clarifying questions above make clear where I was confused.*

The text has been rephrased and is now hopefully clearer. See also our response to comment n°21.

43. *Figure 5: It may help in clearing up my confusion, and perhaps similarly confused readers, if your illustration figures for each strategy showed if each collection of spots were in the same k-cluster or not. Because judging by the contrast in dot colours between the 2nd and 3rd strategies, I would assume that training models were not grouped by k-clusters.*

See our response to comment n°21. In the previous version of Figure 5 (now listed as Figure 6), a color code represented whether 'spots' were in the same k-cluster or not. The color code was indeed poorly chosen and we modified it to make the spots' cluster affiliation more apparent (we also used different symbols). Thank you for pointing this out.

Reviewers' Comments:

Reviewer #1:

Remarks to the Author:

As I wrote in my review of the original manuscript, this is an excellent, important, and much-needed study that reveals central problems with commonly used algorithms for mapping forest biomass and other ecological quantities. The authors have satisfactorily addressed my previous comments. I have only a few suggestions for the authors to consider.

The one response I wasn't happy with concerned the availability of code. I don't think it is adequate to have it available by correspondence from an author. Authors can lose files, fail to respond, and eventually everyone dies. Even if there are other tools available to do these analyses, even if the authors don't feel their code is in great condition to be shared, I think it is a fundamental issue of reproducibility and transparency in this day and age to make the code publicly available somewhere in a format where people can download it without having to contact an author. That could be an appendix, a public github site, or whatever, but it should be available.

I found it very interesting to see the results on the ordinary linear regression in the response to reviews. The authors did not actually include this in the supplemental materials as stated in the response (no doubt an inadvertent error); I recommend including it there along with a bit more explanation of the model, whether in the caption or in another form.

I wonder if one more analysis would highlight the underlying phenomenon that enables models fitted to environmental data to perfectly capture what are really spatially structured patterns. What is the r^2 for predicting X and Y (geographical coordinates) from the suite of environmental variables? I suspect that under random forest it exactly parallels the r^2 for AGB from XY and for AGB from the environmental variables. That is, the reason that these highly flexible models based on many environmental variables can predict AGB is that AGB is spatially structured, and once you have 30-odd environmental variables the combinations of environmental variables are locally sufficiently unique that you can predict location from the environmental variables, and thereby predict the AGB.

Model naming – I wonder if it would be better to rename RF_Full. RF_FULL is contrasted with RF_XY. The full model is usually the model that includes everything, so that would suggest it also includes xy, but of course it doesn't. Maybe RF_ENV instead?

There are quite a number of instances where the wording or phrasing just doesn't sound quite right to a native speaker (too many to enumerate; I mention only a subset in my specific comments). I recommend having the text edited for English before publication.

Other specific comments

The title could in my view be more general. Right now it is focused on the case study here. But the larger point of the manuscript is the pitfalls of the fitting method. That is not in the title at all, really.

Abstract, Lines 34-36. In my view this text is too weak and tentative in its conclusions. Reword to make clear that the apparently high predictive power is not misleading.

L41. Wording.

L47 Wording.

L59-62. Wording can be improved to more directly make these points.

L66-68. Wording

L75-77. Wording. Meaning unclear.

L81-82. And further, the climate and soils datasets themselves are not always very good quality for the tropics...

L78-102. This paragraph is too long. Suggest new paragraph around 95.

Figure 1. For consistency, use grey for forest in the inset map too?

L108-142. Paragraph too long.

L252-264. Caption could be more clear. Wording issues. "Evolution" is the wrong word.

L281. Wording.

L303-305. Wording.

306. "Suspect" is too weak.

L306-350. Paragraph way too long.

L375-77. Wording.

L377-8. References?

Table 1. Suggest including information on the spatial grain of these datasets in the table header. For the topography variables, what is the spatial scale at which the metrics are calculated, and over which they are averaged? For example, slope calculated at 20-m scale is different from that calculated at a 100-m scale for the same landscape, and a 1x1 km value for slope could be the means over the 20-m scale or over 100-m scale values or a 1km-scale value. Similarly for convexity – a point can be on a small hill within a larger valley, and whether its convexity is a valley or ridge value will depend on the spatial scale at which it is calculated.

L476. Remind readers what B-LOO stands for.

For the B-LOO CV, does the amount of training data change as the radius of exclusion changes? If so, then would it make sense to adjust the method slightly to keep the number of training datasets the same? Calculate the number of training data points available at the maximum exclusion radius, and then randomly choose that number of training data points from the potential points for any smaller radii.

Reviewer #2:

Remarks to the Author:

I believe the authors have adequately addressed my comments from my first round of reviews and think this manuscript is of publication quality. This work provides very valuable evidence on the dangers of ignoring spatial autocorrelation in ecological work and I hope it contributes to improving the rigour and reliability of global studies into the future.

RESPONSE TO REVIEWERS

REVIEWER #1.

General comments.

1. As I wrote in my review of the original manuscript, this is an excellent, important, and much-needed study that reveals central problems with commonly used algorithms for mapping forest biomass and other ecological quantities. The authors have satisfactorily addressed my previous comments. I have only a few suggestions for the authors to consider.

Thank you for your positive comments on this study.

2. The one response I wasn't happy with concerned the availability of code. I don't think it is adequate to have it available by correspondence from an author. Authors can lose files, fail to respond, and eventually everyone dies. Even if there are other tools available to do these analyses, even if the authors don't feel their code is in great condition to be shared, I think it is a fundamental issue of reproducibility and transparency in this day and age to make the code publicly available somewhere in a format where people can download it without having to contact an author. That could be an appendix, a public github site, or whatever, but it should be available.

Following your recommendation, we will make our code publicly available. The code will be deposited in the Figshare repository. In practice, we will share the same code we shared with reviewers i.e. the code that allows reproducing the results we obtained with the B-LOO CV, spatial CV and random CV. We will also provide a toy example for users to run the code (just like we did for the review).

3. I found it very interesting to see the results on the ordinary linear regression in the response to reviews. The authors did not actually include this in the supplemental materials as stated in the response (no doubt an inadvertent error); I recommend including it there along with a bit more explanation of the model, whether in the caption or in another form.

Perhaps we did not upload the revised version of the Supplementary material when re-submitting and we apologize for this. The result you are looking for is presented in Supplementary Figure 1. As stated in the main text of the manuscript, the alternative modelling approach we present in Fig. S1 is a "*linear modelling approach*", namely a Partial Least Squares Regression (PLSR, as specified in the figure's legend), and is not a "*ordinary linear regression*". Apart from the modelling approach (i.e. RF vs PLSR), the variables are exactly the same as in the RF model, which is stated in the figure's legend.

4. I wonder if one more analysis would highlight the underlying phenomenon that enables models fitted to environmental data to perfectly capture what are really spatially structured patterns. What is the r^2 for predicting X and Y (geographical coordinates) from the suite of environmental variables? I suspect that under random forest it exactly parallels the r^2 for AGB from XY and for AGB from the environmental variables. That is, the reason that these highly flexible models based on many environmental variables can predict AGB is that AGB is spatially structured, and once you have 30-odd environmental variables the combinations of environmental variables are locally sufficiently unique that you can predict location from the environmental variables, and thereby predict the AGB.

We found this idea to illustrate the problem we address very interesting. It is, however, a bit tricky to test: predicting geographic coordinates requires a modeling approach that handles multivariate responses (i.e. X and Y coordinates), which is not the case of the traditional Random Forest (RF). We did find a multivariate version of RF (MultivariateRandomForest R package). Using this package, the computation time to train a single RF model predicting pixels geographic coordinates based on our set of predictors is about a week when using our full set of observations (c. 30,000 pixels). Looking at how model R^2 varies with the size of the exclusion buffer (and whether it “parallels the R^2 for AGB from XY and for AGB from the environmental variables”, as you suggested) requires training the model c. 16,000 times, it is therefore not feasible (the computation would take years).

Nevertheless, we did run some test on a subset of our observation set, randomly selecting 500 training pixels. Using this subset, we found that pixels geographic coordinates could be predicted with a Root Mean Square Error (RMSE) of c. 25 km on X and c. 15 km on Y when using the full set of predictors. Given the large range of autocorrelation in AGB (> 100 km), this result indeed supports your suspicion that *“the combinations of environmental variables are locally sufficiently unique that you can predict location from the environmental variables, and thereby predict the AGB”*. We therefore included this new result at the end of the Results section:

“An intuitive illustration of this phenomenon can be obtained by taking the reverse approach of predicting geographical coordinates using the optical and environmental predictors. Doing this exercise, we obtained a RMSE on the predicted coordinates of about 25 km. This means that the combinations of predictor variables are sufficiently unique to predict pixel approximate location, and thereby predict the AGB.”

We also added a paragraph to the Method section explaining how this new result was obtained (subsection “Predicting pixel geographic location”).

5. Model naming – I wonder if it would be better to rename RF_Full. RF_FULL is contrasted with RF_XY. The full model is usually the model that includes everything, so that would suggest it also includes xy, but of course it doesn't. Maybe RF_ENV instead?

This is a good point. We modified the name of RF_{FULL} by RF_{RSE} , with the subscript RSE standing for Remote-Sensing and Environmental variables. We found RF_{RSE} more appropriate than RF_{ENV} , since RF_{ENV} would imply that the model only includes environmental variables, when it also includes MODIS reflectance data (besides, environmental data used here derive from – or are extrapolated with – remote-sensing data).

6. There are quite a number of instances where the wording or phrasing just doesn't sound quite right to a native speaker (too many to enumerate; I mention only a subset in my specific comments). I recommend having the text edited for English before publication.

Following your comment, we did have the text edited for English language.

Specific comments

7. The title could in my view be more general. Right now it is focused on the case study here. But the larger point of the manuscript is the pitfalls of the fitting method. That is not in the title at all, really.

We modified the title to: “Overoptimistic nonspatial validation of ecological mapping models exemplified by forest biomass-density maps”. This new title generalizes the point we are making

while still mentioning the subject of the study case.

8. Abstract, Lines 34-36. In my view this text is too weak and tentative in its conclusions. Reword to make clear that the apparently high predictive power is not misleading.

Given the results of our study case, we understand that this conclusion might sound 'too weak'. However, the impact of the methodological flaw we addressed on model validation statistics is study-dependent, in that it will vary with the autocorrelation in the data and with the spatial layout of reference observations (i.e., training & testing data), among other factors. For our conclusion to have a broad validity, we thus decided to remain rather cautious in its formulation. We do not believe that it weakens the warning we send to ecological modelers.

9. L41. Wording.

We rephrased this sentence.

10. L47 Wording.

We rephrased this sentence.

11. L59-62. Wording can be improved to more directly make these points.

We slightly modified this sentence (removing the term 'model' in two occasions). We acknowledge that the last sentence might sound a bit redundant, but we wanted to make the objective of the study perfectly clear. This sentence was added following one of R#2's comment in the last round of review.

12. L66-68. Wording

We rephrased this sentence.

13. L75-77. Wording. Meaning unclear.

We rephrased this sentence.

14. L81-82. And further, the climate and soils datasets themselves are not always very good quality for the tropics...

You are perfectly right. Perhaps all the more in central Africa, where there are – for example – very few climate stations.

15. L78-102. This paragraph is too long. Suggest new paragraph around 95.

We split the paragraph.

16. Figure 1. For consistency, use grey for forest in the inset map too?

Done

17. L108-142. Paragraph too long.

We split the paragraph.

18. L252-264. Caption could be more clear. Wording issues. “Evolution” is the wrong word.

We rephrased the caption and removed the term ‘evolution’.

19. L281. Wording.

We rephrased this sentence.

30. L303-305. Wording.

We rephrased this sentence.

31. 306. “Suspect” is too weak.

We rephrased this sentence.

32. L306-350. Paragraph way too long.

We split the paragraph in three.

33. L375-77. Wording.

We rephrased this sentence.

34. L377-8. References?

We added a reference.

35. Table 1. Suggest including information on the spatial grain of these datasets in the table header. For the topography variables, what is the spatial scale at which the metrics are calculated, and over which they are averaged? For example, slope calculated at 20-m scale is different from that calculated at a 100-m scale for the same landscape, and a 1x1 km value for slope could be the means over the 20-m scale or over 100-m scale values or a 1km-scale value. Similarly for convexity – a point can be on a small hill within a larger valley, and whether its convexity is a valley or ridge value will depend on the spatial scale at which it is calculated.

Following a similar comment in the last round of review, we added this information in the text of the Method section. Unfortunately, we cannot add one supplementary column to Table 1 while complying with editorial guidelines on tables formatting (the table width would be too large to fit in a single page). We, however, added a footnote to the table, reminding readers about the spatial grain and extent over which topography variables were computed.

36. L476. Remind readers what B-LOO stands for.

Done

37. For the B-LOO CV, does the amount of training data change as the radius of exclusion changes? If so, then would it make sense to adjust the method slightly to keep the number of training datasets the same? Calculate the number of training data points available at the maximum exclusion radius, and then randomly choose that number of training data points from the potential points for any smaller radii.

Yes, the amount of training data decreases as the exclusion radius around a test observation increases. For the largest exclusion radius ($r = 150$ km), the size of the training dataset decrease by 15.3 ± 9.8 % on average. The methodological point you are making in this comment is relevant, we thus modified our B-LOO CV method following your proposition. After a week of computation, we processed 700 test cells using the updated B-LOO CV (out of the 1000 tests cells used in the previous version of the results). The updated results were similar to the old ones (see Fig. 1 below). We thus left the original version of Fig. 5 in the main manuscript.

Figure 1. Results of the updated B-LOO CV procedure on 700 test cells (corresponding to Fig. 5 – a in the main manuscript).

REVIEWER #2.

General comments.

1. I believe the authors have adequately addressed my comments from my first round of reviews and think this manuscript is of publication quality. This work provides very valuable evidence on the dangers of ignoring spatial autocorrelation in ecological work and I hope it contributes to improving the rigour and reliability of global studies into the future.

Thank you for your help improving the manuscript.